# Explaining Concept Shift with Interpretable Feature Attribution

## Abstract

Regardless the amount of data a machine learning (ML) model is trained on, there will inevitably be data that differs from their training set, lowering model performance. Concept shift occurs when the distribution of labels conditioned on the features changes, making even a well-tuned ML model to have learned a fundamentally incorrect representation. Identifying these shifted features provides unique insight into how one dataset differs from another, considering the difference may be across a scientifically relevant dimension, such as time, disease status, population, etc. In this paper, we propose SGShift, a model for detecting concept shift in tabular data and attributing reduced model performance to a sparse set of shifted features. We frame concept shift as a feature selection task to learn the features that can explain performance differences between models in the source and target domain. This framework enables SGShift to adapt powerful statistical tools such as generalized additive models, knockoffs, and absorption towards identifying these shifted features. We conduct extensive experiments in synthetic and real data across various ML models and find SGShift can identify shifted features with AUC > 0.9, much higher than baseline methods, requires few samples in the shifted domain, and is robust in complex cases of concept shift. Applying SGShift to 2 real world cases in healthcare and genetics yielded new feature-level explanations of concept shift, including respiratory failure's reduced impact on COVID-19 severity after Omicron and European-specific rare variants' impact on Lupus prevalence.

## 1 Introduction

Machine learning (ML) models are often trained on vast amounts of data, but will inevitably encounter test distributions that differ from the training set. Such distribution shift is one of the most common failure modes for ML in practice. When models do fail, model developers need to diagnose and correct the problem. In the simplest case, this may simply consist of gathering more data to retrain the model. However, in other cases, it may be necessary to fix issues in an an underlying data pipeline, add new features to replace ones that have become uninformative, or undertake other more complex interventions. A necessary starting point for any such process is to understand what changed in the new dataset. Developing such understanding may even have scientific importance. For instance, a novel virus variant may emerge with new risk factors, lowering the performance of models that predict disease progression, or specific mutations in the genome could have differing relevance to disease between ancestries, weakening polygenic risk score models due to fundamentally different biology between populations (Duncan et al., 2019; Martin et al., 2019).

We propose methods for diagnosing distribution shift, focusing specifically on the case of *concept shift*, or when the conditional distribution of the label given the features, $p(y|X)$, differs between the source and target distribution. Concept shift represents the difficult case where the relationship between features and outcome has changed, as opposed to marginal shifts impacting only $X$ or $y$ by themselves. Indeed, (Liu et al.,

2024) document concept shift as the primary contributor to performance degradation across a wide range of empirical examples of distribution shifts. In this setting, our goal is to understand how $p(y|X)$ differs between the source and target domains.

Understanding distribution shift has been the subject of increasing interest. However, existing methods mostly operate relative to a structure for the data which is prespecified by the analyst, for example a known causal graph (Zhang et al., 2023; Subbaswamy et al., 2021), fixed decomposition of the variables (Singh et al., 2024), or particular assumed models for distribution shift in which out-of-distribution performance can be identified using only unlabeled data Chen et al. (2022). Methods that do not impose such structural conditions largely repurpose other tools to explain distribution shift as a secondary objective. For example Mougan et al. (2023) propose to look for changes in model explanations, while Liu et al. (2023) fit a decision tree to explain differences in predictions from source and target domain models as part of a larger empirical investigation.

We introduce SGShift, a new method directly designed for diagnosing distribution shift. SGShift offers robust statistical performance, particularly with limited target-domain samples and without requiring prespecified causal structure. Just as sparsity is an effective principle for learning predictive models in many settings due to sparse mechanism shift (Schölkopf et al., 2021), we hypothesize that the *update* to $p(y|X)$ between the source and target domains may often be sparse (a fact that we empirically verify in several application domains). In this case, a useful explanation of concept shift is to identify a small set of features that drive the change between the two distributions, which could e.g. be the subject of potential modeling fixes. SGShift frames this problem as learning an update to a source distribution's predictive model using a minimal set of features to recover the performance loss in the target distribution. We show how this formulation allows simple, principled, and easily implemented diagnoses of distribution shift, without requiring any prior knowledge, causal information, or parametric priors regarding the dataset.

We benchmark SGShift against several baselines on semi-synthetic datasets with known feature shifts, observing greatly superior performance at identifying concept shifted features (referred to as shifted features throughout). We then apply SGShift to two real-data settings and recover real-world concept shifts consistent with findings from medical and biological literature, such as respiratory failure's reduced impact on COVID-19 severity after Omicron and European-specific rare variants' impact on Lupus prevalence. Together, these findings provide evidence that SGShift can recover accurate and interpretable descriptions of distribution shift across a wide range of settings.

## 1.1 Additional related Work

**Covariate shift.** Much of the existing work on distribution shift has focused on detecting or correcting shifts in the marginal feature distribution, $P(X)$, e.g. covariate shift with the assumption that $P(y|X)$ remains unchanged. For instance, (Kulinski et al., 2020) introduce statistical tests to identify which variables have shifted between source and target domains, while (Kulinski & Inouye, 2023) propose explaining observed shifts via a learned transportation map between the source and target distributions, not distinguishing between features and labels. $P(X)$ shift can be identified by methods like two-sample tests (Jang et al., 2022) or classifiers (Lipton et al., 2018) and corrected by techniques such as importance sampling (Sugiyama et al., 2007). Cai et al. (2023) further use these ideas to correct covariate shift by regarding the unexplained residual as a shift in $P(y|X)$, although they don't correct or explain the concept shift. Although these methods can be effective for addressing covariate shift, they often do not delve into potential shifts in the conditional distribution. Explaining shifts in $P(y|X)$ typically involves performing feature-by-feature analyses of the conditional distribution $P(y|X_i)$ (Guidotti et al., 2018). However, such univariate assessments risk detecting spurious shifts due to unadjusted confounding in the presence of collinearity among predictors (Raskutti et al., 2010). Kulinski & Inouye (2023) consider an unsupervised setting where the goal is to identify a set of features whose distribution differs (e.g., sensors that have been compromised by an adversary), as opposed to identifying features whose relationship with a supervised label has changed.

**Conditional distribution shift.** Recent efforts have begun to tackle shifts in the conditional distribution $P(y|X)$ more directly. For example, (Zhang et al., 2023) consider changes in a causal parent set as a whole, relying on known causal structures. (Mougan et al., 2023) propose a model-agnostic "explanation shift detector" that applies SHAP (Shapley additive explanations) to a source-trained model and covariates in both source and target domains, without including the outcomes in the target domain. They then use a two-sample test on the feature-attribution distributions from SHAP to detect whether the model's decision logic has changed because of the changing of $P(X)$ across domains. Despite its effectiveness in signaling shifts, this approach does not pinpoint which features are driving the changes in $P(y|X)$. (Singh et al., 2024) decompose the domain loss gap into predefined marginal and conditional segments, then allocate feature-level contributions, while (Singh et al., 2025) automatically discover subgroups within the data for which to produce feature-level explanations. (Subbaswamy et al., 2021) stress tests a source model before distribution shift, requiring a prespecified set of shifting variables. (Chen et al., 2022) focus on estimation of performance shift on an unlabeled dataset, but this require restrictive assumptions for identifiability, particularly that non-shifted features have no shifts at all when conditioned on the shifted features and label between datasets. WhyShift (Liu et al., 2023) compares two independently trained models - one from each domain - and analyze their difference to locate regions of covariate space with the largest predictive discrepancy. SGShift differs in that we aim to explicitly identify what the features contributing to conditional distribution shift are without requiring any prior knowledge of the dataset.

## 2 PRELIMINARIES AND PROBLEM FORMULATION

**ML prediction tasks.** We consider standard ML tasks, such as classification, regression, etc. Given features $X \in \mathcal{X} \subseteq \mathbb{R}^p$, the goal is to predict associated labels $y \in \mathcal{Y}$. Let $h(\cdot)$ denote an ML model applicable to this task. Given this model's predictions $h(X)$ and true labels $y$, the performance can be quantified by a loss $\ell(\hat{y}, y)$. This can be any loss, such as 0-1 loss in classification or MSE in regression.

**Conditional distribution shift.** ML models are typically trained on one set of data, and then applied to another. This training and inference data often come from different distributions, referred to as source and target domains. We consider the particular case of *conditional distribution shift*, where the probability of observing $y$ given the same $X$ differs between source and target domains. Formally, let $P_S$ and $P_T$ denote the probability density/mass function of the source and target domains, respectively. Conditional distribution shift occurs when $P_S(y \mid X) \neq P_T(y \mid X)$.

**Problem formulation.** We consider the problem of identifying the set of features that cause conditional distribution shift. Suppose we observe i.i.d. samples $(X_i^{(S)}, y_i^{(S)})_{i=1}^{n_S} \sim P_S$ and $(X_i^{(T)}, y_i^{(T)})_{i=1}^{n_T} \sim P_T$, where $n_S$ and $n_T$ are the number of samples in source and target domain. A source model $h_S(\cdot)$ is trained and applied to the target domain. A shift happens such that $P_T(y \mid X) \neq P_S(y \mid X)$ for at least one feature in $X$, thus $h_S(\cdot)$ underperforms when applied to $T$. Our goal is to identify the smallest set of shifted features $A \in X$ on which the change depends. Formally, consider the difference between the conditional expectation functions,

$$\Delta(X) = d'(\mathbb{E}_S[y|X], \mathbb{E}_T[y|X]).$$

for some difference metric $d'$. In some cases, we may also choose to model $\Delta$ on a transformed scale, e.g., the logit scale for a binary response, in which case we will take $\Delta(X) = g(\mathbb{E}_S[y|X]) - g(\mathbb{E}_T[y|X])$ for some link function $g$. Our hypothesis is that for many realistic distribution shifts, $\Delta$ will be (approximately) sparse, i.e., depending on only a small number of inputs in $X$. Let $A \subseteq X$ denote this support set. For example, this may be the case if specific nodes in a causal process generating the data are intervened on, as is the premise for several previous models of distribution shift (Chen et al., 2022) as well as the concept of sparse mechanism shift in causal representation learning (Schölkopf et al., 2021). Our goal is to recover the support set $A$ to serve as an explanation of the shift. In practice, we may not expect that sparsity is exactly satisfied,

so we look for a $\Delta$ that solves

$$\min_{\hat{\Delta}} d(\Delta(X), \hat{\Delta}(X)) \quad \text{s.t. } \hat{\Delta} \text{ is } k\text{-sparse}$$

for some distance function $d$. $k$-sparse denotes that $\Delta$ is constant with respect to all but $k$ inputs, and we search across a range of values of $k$ to identify a level of sparsity at which $\Delta$ is well-approximated.

The problem is potentially challenging because $\Delta$ is the difference between two regression functions over different data distributions. For any given training point $X$, we see either a label $y$ from distribution $S$ or distribution $T$, but never both. Accordingly, it is not possible to directly apply existing methods for sparse regression. The most directly related work, the WhyShift framework introduced by (Liu et al., 2023) for diagnosing concept shift, takes a plugin approach. A plugin strategy first fits models on the two separately datasets to approximate $\mathbb{E}_S[y|X]$ and $\mathbb{E}_T[y|X]$. Second, it fits a second model regressing some difference metric of $\widehat{\mathbb{E}}_S[y|X]$ and $\widehat{\mathbb{E}}_T[y|X]$ on $X$ to summarize the structure in $\Delta$. However, this plugin approach risks an accumulation of errors, particularly when we are interested in recovering structure related to sparsity: given noisy approximations to the two conditional expectations, the difference between $\widehat{\mathbb{E}}_S[y|X]$ and $\widehat{\mathbb{E}}_T[y|X]$ will not necessarily display the same sparsity pattern as $\Delta$ (as we observe experimentally). It is also potentially challenging when we have limited target-domain data, since separately fitting $\mathbb{E}_T[y|X]$ may be especially difficult in this setting.

## 3 METHOD

Our method, SGShift, circumvents these difficulties by reformulating the above problem in a way that allows existing sparse regression methods to be applied in a black-box fashion. Instead of first fitting separate models for $\mathbb{E}_S[y|X]$ and $\mathbb{E}_T[y|X]$ and then finally using them to fit $\Delta$, SGShift starts with just a source-distribution model $h_S(X)$. We then find a sparse *correction* term such that the corrected model has maximum target-distribution performance. Formally, SGShift solves

$$\min_{\hat{\Delta}} \mathbb{E}_T[\ell(h_S(X) + \hat{\Delta}(X), y)] \quad \text{s.t. } \hat{\Delta} \text{ is } k\text{-sparse}$$

This recipe has two advantages. First, it can be instantiated with any sparse regression method, taking the source-distribution model $h_S(X)$ as a fixed "constant" term that is applied to each sample. Second, we can separately control the complexity of the model used for the source vs correction term: when source-domain data is abundant, $h$ may be relatively complex, but under the common challenge of limited target-domain data, we can use a simpler model for $\hat{\Delta}$. In this work, we instantiate SGShift using $\ell_1$ regularization for sparsity and knockoffs for false discovery control, as these are widely used, easy to implement, and tend to perform robustly in practice. We show that SGShift directly inherits the theoretical guarantees of these methods for recovery of the support set, despite the fact that the outcome we are attempting to recovery the sparsity pattern for is never directly observed. However, other sparse regression methods can be applied out-of-the-box to fit the characteristics of specific data distributions.

### 3.1 SGSHIFT: INSTANTIATION WITH $\ell_1$ REGULARIZATION

Our suggested implementation of SGShift uses a generalized additive model (GAM) with $\ell_1$ regularization to model the correction term. Specifically, we model

$$g(\mathbb{E}_T[y \mid X]) = h_S(X) + \phi(X)^\top \delta \tag{1}$$

where $g$ is a link function, $\phi(X)$ is a set of basis functions chosen by the user (by default, $\phi(X) = X$), $\delta$ is a vector of coefficients for the correction term. The GAM link function $g$ allows the user to model sparsity

on, e.g., the logits scale when $y$ is binary, which may be more natural than the probability scale. In order to control the sparsity level of $\delta$, SGShift imposes $\ell_1$ regularization and solve

$$\hat{\delta} = \arg \min_{\delta \in \mathbb{R}^K} \left\{ L(\delta) + \lambda \|\delta\|_1 \right\} \quad L(\delta) := \ell(h_S(X_T) + \phi(X_T)^\top \delta, y_T) \tag{2}$$

where $\ell$ here is the negative log-likelihood for the generalized additive model and $\lambda$ is a regularization parameter which we vary to obtain solutions of a range of sparsity levels.

## 3.2 SGShift-A: Refined fitting considering source model misspecification

Prioritizing shifted features relies on an existing model trained on the source dataset. However, it may be that this model does not represent the data well due to difficulties in model fitting. To avoid source model misfit biasing the selection of shifted features, we incorporate an additional absorption term to nullify this effect. The main absorption idea is that the error from fitting occurs in both domains, while the conditional shift occurs only in the target domain. We solve:

$$(\hat{\omega}, \hat{\delta}) = \arg \min_{\omega, \delta \in \mathbb{R}^K} \left\{ \ell\left( \underbrace{\begin{bmatrix} h_S(X_S) \\ h_S(X_T) \end{bmatrix}}_{\text{offset}} + \underbrace{\begin{bmatrix} \phi_S^\top & \mathbf{0} \\ \phi_T^\top & \phi_T^\top \end{bmatrix} \begin{bmatrix} \omega \\ \delta \end{bmatrix}}_{\text{absorption}}, \begin{bmatrix} y_S \\ y_T \end{bmatrix} \right) + \lambda_\omega \|\omega\|_1 + \lambda_\delta \|\delta\|_1 \right\} \tag{3}$$

where $\phi_S$ and $\phi_T$ refer to the values of basis functions in source and target domains, $\omega \in \mathbb{R}^K$ acts on both domains and $\delta \in \mathbb{R}^K$ is in the target domain only. We induce hierarchical regularization $\lambda_\omega < \lambda_\delta$ to penalize the inference of shift more heavily than model misspecification to be conservative in identifying shifted features.

## 3.3 SGShift-K: Explicit false discovery control with knockoffs

While $\ell_1$ regularization enables recovery of a sparse correction vector $\delta$, we may wish for principled guarantees that limit the false discovery of features that did not in fact shift. For this purpose, we adapt the knockoffs framework (Candes et al., 2018). Knockoffs generate synthetic features that mimic the correlation structure of the real data to limit false discoveries. Following (Candes et al., 2018), we construct a Model-X knockoff matrix $\tilde{X} = [\tilde{X}^{(1)}, \dots, \tilde{X}^{(p)}] \in \mathbb{R}^{n \times p}$ and apply SGShift's variable selection procedure to the basis-transformed design matrix $[\phi \ \tilde{\phi}] = [\phi(X) \ \phi(\tilde{X})] \in \mathbb{R}^{n \times 2K}$. We then form a combined coefficient vector $\delta' = \begin{bmatrix} \delta \\ \tilde{\delta} \end{bmatrix} \in \mathbb{R}^{2K}$ where $\delta$ corresponds to original basis functions and $\tilde{\delta}$ to their knockoffs. The details of the construction and selection with knockoffs is in Appendix C. Unlike classical knockoff regression, however, our model is applied not to the raw features but to the *additive correction term* on top of the predictive model trained on source domain $\hat{f}(X_T)$. Concretely, we treat $\hat{f}(X_T)$ as a fixed offset and fit the residual correction using both original and knockoff basis functions. The optimization problem becomes:

$$\hat{\delta}' = \arg \min_{\delta' \in \mathbb{R}^{2K}} \left\{ \ell\left( h_S(X_T) + [\phi_T \ \tilde{\phi}_T]^\top \delta', y_T \right) + \lambda \|\delta'\|_1 \right\}. \tag{4}$$

We then apply the standard derandomized knockoffs procedure for feature selection (Ren et al., 2023), which effectively uses the knockoff features – that are known to be "fake" – to set a threshold for inclusion in the returned set. Notably, the objective of SGShift-K is shifted feature selection only with the generation of knockoff copies, while SGShift and SGShift-A can do simultaneous feature selection and target model correcting from the trained source model.

## 3.4 THEORETICAL GUARANTEES

We show that when the model in Equation 1 is well-specified, SGShift has desirable theoretical guarantees on recovery of the true shift coefficients $\delta$ under proper choice of the regularization parameter $\lambda$. Importantly, this only requires imposing assumptions on the form of the between-distribution difference $\Delta$, rather than on the complete regression function $\mathbb{E}_T[y|X]$, which is allowed to be nonparametric (as opposed to the standard Lasso setting). In particular, we obtain the following:

**Theorem 3.1** (Convergence Guarantee for $\delta$ from SGShift (with Equation 2)). *Assume $\delta^* \in \mathbb{R}^K$ be the true parameter with support $A \subseteq [K]$, $|A| = a$, $\phi(X)$ be sub-Gaussian. Suppose (1) Loss function $L$ satisfies Restricted Strong Convexity (RSC, justification in Appendix A) (2) Subgradient Bound: $\|\nabla L(\delta^*)\|_\infty \lesssim \lambda$. (3) Regularization Parameter: $n_T \lambda = \lambda' \asymp \sqrt{\log K / n_T}$. Then, with probability approaching 1, the estimation error $\hat{\delta} - \delta^*$ satisfies $\|\hat{\delta} - \delta^*\|_2^2 \lesssim \frac{a \log K}{n_T}$.*

The proof is in Appendix B. Here, $\lesssim$ means asymptotically bounded above up to a constant factor, and $\asymp$ means asymptotically the same order up to constant factors.

Further, the use of knockoffs in the second stage allows us to prove stronger guarantees on the probability that any feature is false included in the selected set.

**Theorem 3.2** (Stability Selection Control). *Let $A^c = \{k : \delta_k^* = 0\}$ denote the set of features with zero coefficient in the true data distribution and $B$ the number of knockoff samples.*

*(PFER Control) Assume for each $k \in A^c$, $P(k \in \hat{A}^{[b]}) \leq \alpha$ uniformly over b, where $\alpha$ is the per-iteration false selection probability controlled via $\tau$ and $\hat{A}^{[b]}$ is the estimated A for bth knockoff repeat, and $\hat{A}(\pi)$ is the selection across all repeats under stability threshold $\pi$. For any stability threshold $\pi > \alpha$:*

$$\mathbb{E}\left[|\hat{A}(\pi) \cap A^c|\right] \leq |A^c| \exp\left(-2B(\pi - \alpha)^2\right).$$

*(FDR Control) Assume each $\hat{A}^{[b]}$ satisfies $\mathbb{E}\left[\frac{|\hat{A}^{[b]} \cap A^c|}{|\hat{A}^{[b]}| \vee 1}\right] \leq q$ (FDR control at level q via $\tau$) as per Theorem 3.1 in (Candes et al., 2018). Then:*

$$FDR(\hat{A}(\pi)) \leq \frac{q}{1 - (1 - \pi)^B}.$$

Theorem 3.2 guarantees both per family error rate (PFER) and false discovery rate (FDR) control under proper parameter selection. The proof is in Appendix D. We also provide a discussion of parameter selection for FDR control for SGShift in Appendix E.

## 4 EXPERIMENTS

**Evaluation setup** We evaluate our method on three real-world healthcare datasets (details in Appendix F) exhibiting natural distribution shifts, 30-day Diabetes Readmission (Strack et al., 2014) split by ER admission, COVID-19 Hospitalizations (of Us Research Program Investigators, 2019) split by pre and post-Omicron, and SUPPORT2 Hospital Expenses (Connors et al., 1995) split by death in hospital. For each of these 3 naturally shifted datasets, we construct semi-synthetic simulations, consistent with previous work (Singh et al., 2025; Zhang et al., 2023). We fit a "generator" model" to the real labels in source domain, relabeling the source data, then simulate the target dataset's labels with an induced conditional shift by perturbing $g(E[y|X])$ based on selected input features. A "base" model is then trained from the relabeled source domain. We vary base and generator models to be each combination of decision tree, logistic/linear regression, gradient boosting, and support-vector machines, for a total of 16 settings in each dataset and 48 total settings. We consider 4

scenarios in each setting, sparse shift, where a small set of features are shifted, dense shift, where >60% of the features are shifted, global shift, where all features shift slightly, with a few shifting greater than others, and interaction shifts, occurring in the interaction space. All features to shift are selected randomly. We additionally consider high dimensional, highly correlated, and low signal-to-noise simulation settings. In feature selection tasks, we primarily use SGShift-K with knockoffs, and in model performance recovery we use naive SGShift and SGShift-A with absorption. SGShift's feature ranking is obtained by varying the penalty parameter from 0.0001 to 100 to measure AUC and recall. Full preprocessing details and replication code is in the appendix.

**Baselines** We consider 3 baseline models which also use both features and labels in source and target domain to identify shifted features. **Diff**, a method we construct where we simply compute the outcome discrepancies of two "base models" separately trained on source and target data, and apply sparse regression on held-out samples and the base models' outcome probability differences to identify features contributing to the shifts. **WhyShift** (Liu et al., 2023) uses two "base models" separately trained on source and target domains and computes model outcome probability discrepancies, then trains a non-linear decision tree on these discrepancies to detect regions (paths in the tree) responsible for conditional shifts. We extract the features from any path in the learned tree with feature importance $> 0$ and consider them as the shifted features. **SHAP**, a Shapley value-based method we adapt from (Mougan et al., 2023) such that we can find individual features that differ between datasets. SHAP trains "base models" separately on source and target data, computes the Shapley value of each feature, and ranks the largest absolute differences between models.

### 4.1 BENCHMARKING

**Accuracy in identifying shifted features.** First, we examine the case of sparse shifts, in line with SGShift's sparsity assumption. 5 features are perturbed in each dataset between domains, while the rest remains fixed. Table 1 shows evaluation of SGShift in detecting shifted features in these simulations, measured in AUC at detecting true shifted features (a binary 0/1 label). Across model settings and datasets, SGShift achieves the strongest performance compared to baselines Diff, WhyShift and SHAP, with AUC typically greater than 0.9, 0.1-0.2 higher than the nearest baseline. Despite the presence of model mismatch, SGShift still attains high performance in the mismatched setting, on average only 0.02 AUC below the matched setting.

We next examine the case of dense concept shift, violating SGShift's sparsity assumption. 25/33, 25/30, and 40/64 features are perturbed in each simulation setting of Diabetes readmission, COVID-19, and SUPPORT2 respectively. Table 1 shows evaluation results. Despite the assumption of sparsity, SGShift still attains AUC greater than 0.8 and 0.9. This is in contrast to baseline methods, whose performance may reduce substantially, such as all methods in the Diabetes dataset, each with AUC around 0.6, down from around around 0.75 previously. SGShift is robust towards dense shift and does not over-emphasize a few features when many may be shifted. While perhaps counterintuitive given the sparsity assumption, SGShift likely performs well as it effectively acts a a regularized feature-ranking procedure, and can still capture most of the signal even when most shift coefficients are nonzero. Ablations confirming the utility of SGShift-K over naive SGShift and SGShift-A are in Appendix H.

Next, we vary the sample size available in the target domain, simulating an online learning setting where data is gradually streaming in. Results for COVID-19 are reported in Figure 1. SGShift is able to identify over half the shifting features given only 100 samples, and over 85% given 500 samples. This indicates SGShift is indeed an effective diagnostic tool, not requiring many samples for identifying features or correcting models. Similar results are reported for Diabetes and COVID-19 in Appendix I.

Results of global, interaction, high dimensional, highly correlated, and noisy shifts are available in the Appendix J, K, L, M, P, respectively.

| Model Match | Sparse simulations | | | | Dense simulations | | | |
|---|---|---|---|---|---|---|---|---|
| | Diff | WhyShift | SHAP | SGShift-K | Diff | WhyShift | SHAP | SGShift-K |
| **Diabetes Readmission** | | | | | | | | |
| Matched | $0.64 \pm 0.09$ | $0.73 \pm 0.08$ | $0.77 \pm 0.12$ | $\mathbf{0.90} \pm \mathbf{0.01}$ | $0.54 \pm 0.09$ | $0.52 \pm 0.08$ | $0.64 \pm 0.12$ | $\mathbf{0.86} \pm \mathbf{0.01}$ |
| Mismatched | $0.69 \pm 0.06$ | $0.72 \pm 0.04$ | $0.76 \pm 0.04$ | $\mathbf{0.86} \pm \mathbf{0.04}$ | $0.58 \pm 0.06$ | $0.57 \pm 0.04$ | $0.60 \pm 0.04$ | $\mathbf{0.82} \pm \mathbf{0.04}$ |
| **COVID-19** | | | | | | | | |
| Matched | $0.78 \pm 0.05$ | $0.76 \pm 0.06$ | $0.81 \pm 0.10$ | $\mathbf{0.99} \pm \mathbf{0.02}$ | $0.79 \pm 0.05$ | $0.65 \pm 0.06$ | $0.86 \pm 0.10$ | $\mathbf{0.95} \pm \mathbf{0.02}$ |
| Mismatched | $0.77 \pm 0.03$ | $0.71 \pm 0.05$ | $0.77 \pm 0.03$ | $\mathbf{0.97} \pm \mathbf{0.03}$ | $0.78 \pm 0.03$ | $0.74 \pm 0.05$ | $0.78 \pm 0.03$ | $\mathbf{0.93} \pm \mathbf{0.03}$ |
| **SUPPORT2** | | | | | | | | |
| Matched | $0.83 \pm 0.05$ | $0.67 \pm 0.06$ | $0.82 \pm 0.09$ | $\mathbf{0.96} \pm \mathbf{0.01}$ | $0.62 \pm 0.05$ | $0.56 \pm 0.06$ | $0.62 \pm 0.09$ | $\mathbf{0.92} \pm \mathbf{0.01}$ |
| Mismatched | $0.80 \pm 0.03$ | $0.67 \pm 0.03$ | $0.76 \pm 0.05$ | $\mathbf{0.95} \pm \mathbf{0.01}$ | $0.73 \pm 0.03$ | $0.60 \pm 0.03$ | $0.70 \pm 0.05$ | $\mathbf{0.92} \pm \mathbf{0.01}$ |

Table 1: **Performance in identifying shifted features.** AUC of detecting the true set of shifted features in sparse (left) and dense (right) semi-synthetic simulations. Matched refers to when generator and base model are the same, mismatched when they differ. Results are aggregated across the 4 matched and 12 mismatched settings. 95% confidence intervals are evaluated across configurations.

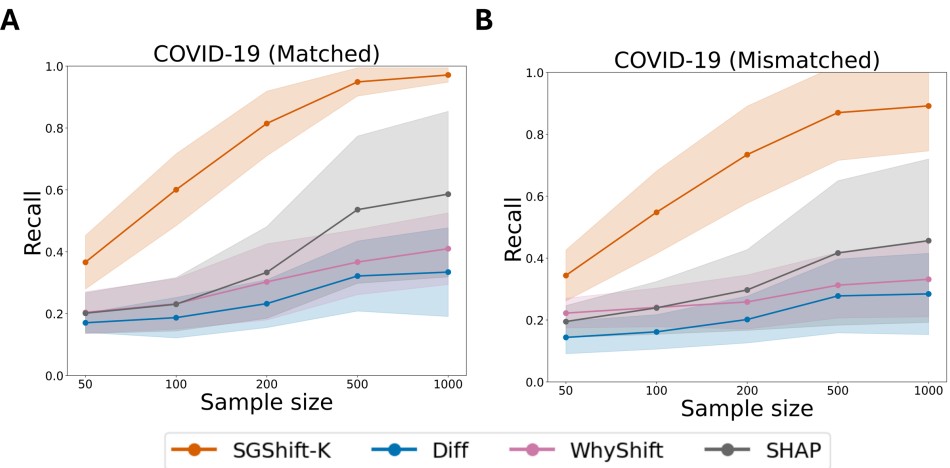

Figure 1: **Performance across sample sizes.** Sample size is varied from 50 to 1000. 95% CI's are shown across 16 simulation settings. Recall is measured at fixed FPR 5%.

## 4.2 REAL DATA

**Real-world sparse concept shift.** We verify the sparsity of true concept shift in Figure 2A. Across datasets and model configurations, SGShift is able to learn updates to the source model that recover 90% of the performance loss in the target domain, requiring less than 1/3 of the total features, and in some cases as little as one feature. As an illustrative example, we show how SGShift recovers performance for a gradient boosting model in the COVID-19 dataset in Figure 2B. With only 1 feature, the performance loss can be completely recovered. Furthermore, adding additional features beyond what is needed may even reduce performance. These results indicate that true concept shift can indeed often be explained by a subset of features shifting. We additionally perform diagnostics to ensure these shifts are not the result of covariate shift in Appendix N. We further analyze another 87 cases of real world concept shift and find that 78 of these can have model

performance completely recovered with less than 30% of the features in Appendix O, indicating sparsity is indeed common in concept shift. Results of additional model configurations and datasets are in the Appendix.

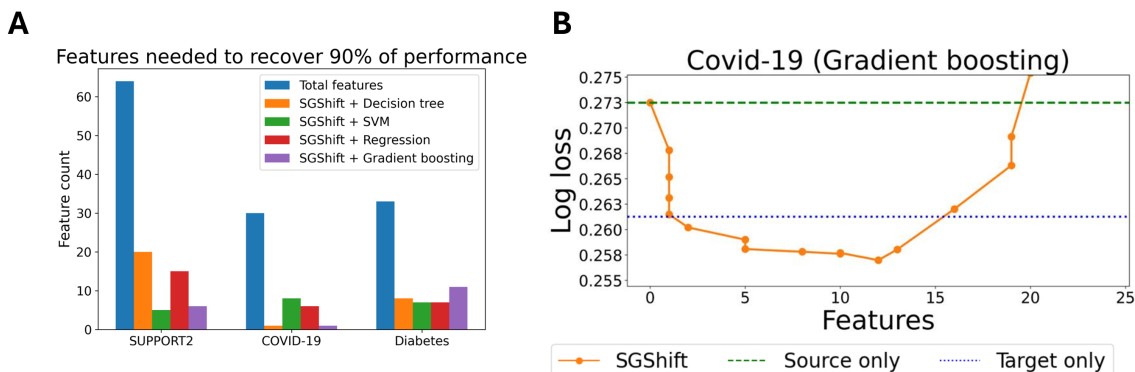

Figure 2: **Sparsity in real world concept shift.** A) How many features SGShift required to learn an update to the source model that recovered 90% of the performance loss in the target domain. B) By decreasing the feature penalization penalty to add more features to SGShift's update, we see how many terms are needed to recover performance in the target domain.

**Case study in healthcare.** We next evaluate the validity of the top features selected by SGShift-K contributing to the shift in COVID-19 severity after Omicron in Figure 3 (data split in Table 2). The highest ranked feature across all models is respiratory failure with a negative sign, consistent with the broad observation of less severity during Omicron compared to the previous Delta variant (Adjei et al., 2022), partly due to Omicron's decreased ability at infecting lung cells (Hoffmann et al., 2023). More severe cases may be taking place in other pathways, such as the upper respiratory tract (Wickenhagen et al., 2025). Abnormal breathing and other circulatory/respiratory signs have decreased risk, likely for the same reason. Non-lung related comorbidities tend to contribute more to increased hospitalization risk, as with decreased lung comorbidity risk, they may now be more relevant to severity (Lewnard et al., 2022).

**Case study in genetics.** We consider a known case of concept shift in the difference in Lupus severity and prevalence between ancestries. We use the gene expression from 149 healthy and Lupus-affected Europeans, and 107 healthy and Lupus-affected Asians (Perez et al., 2022), and aim to predict Lupus status using the top 1000 variable genes in B cells, a cell type commonly implicated in Lupus. We split by ancestry and apply SGShift-K to find genes contributing to concept shift. Expectedly, we first observe an XGBoost model trained on Europeans underperforms when applied to Asians (European AUC 1.0, Asian AUC 0.84. SGShift-K discovers 6 genes in B cells contributing to this shift: ERRFI1, RP11-666A1.5, CTD-2561B21.11, AC012309.5, AC074212.5, and AP001059.5, all with negative coefficients, and completely recovers the performance drop. ERRFI1 and RP11-666A1.5 are both differentially expressed in B cells between these ancestries (Wang & Gazal, 2023). A genetic basis of difference in Lupus between ancestries has been discovered, and CTD-2561B21.11, AC012309.5, AC074212.5, and AP001059.5 are underpinned by eQTLs or repeat variants common in Europeans but rare in East Asians (Morris et al., 2016; Langefeld et al., 2017). Interferon signatures commonly correlate with Lupus prevalence, and Asians have elevated background interferon levels compared to Europeans, such as RP11-666A1.5 (Rector et al., 2023). These results indicate SGShift is picking up true biology underlying the difference in Lupus between European and Asian populations, although we acknowledge these findings would need further validation to ensure results are robust to the many hidden confounders present in biological data.

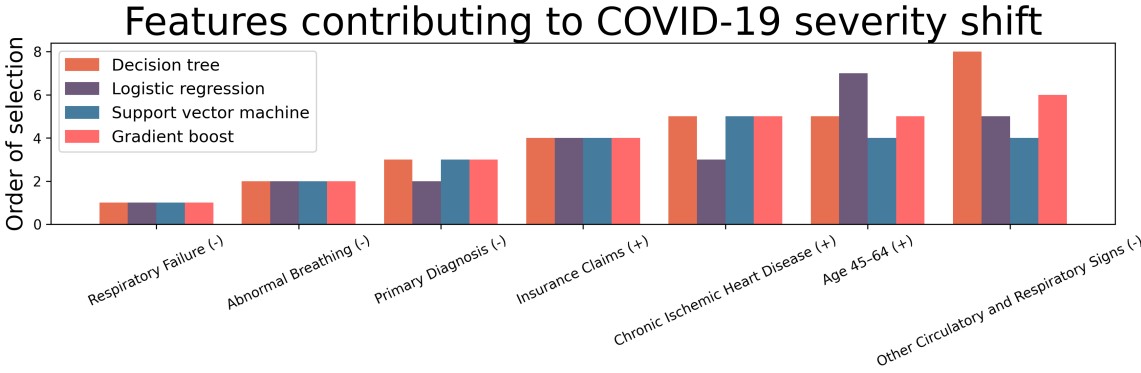

Figure 3: **Shifted features in COVID-19 severity.** Real data results showing the ordering of selected features for each model as the penalty term increases for COVID-19 severity. Positive (+) and negative (-) coefficients are treated as 2 distinct features. Only features selected in the top 5 for any model are shown.

## 5  DISCUSSION

We have presented SGShift, a method for attributing concept shift between datasets to a sparse set of features. Our work contributes towards understanding what makes models fail between datasets. We prove statistical guarantees regarding SGShift's false discovery control and demonstrate high power in detecting true shifted features, even when the assumption of sparsity is violated. We show that true concept shifts in tabular healthcare data do indeed tend to be sparse and SGShift can explain these shifts. Future work could include optimizing model performance by explicitly modeling the difference between datasets given the identified shifted features, disentangling various contributors to concept shift such as label or measurement drift, or extending SGShift to non-tabular data, e.g., images or graphs.

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

## A  JUSTIFICATION OF RESTRICTED STRONG CONVEXITY (RSC)

The RSC condition is central to ensuring the quadric growth of the loss difference around the true parameter $\delta^*$, even in high dimensions. We formalize its validity as follows:

**Lemma A.1** (RSC for $\ell_1$-Penalized Loss). *Let $\phi_i \in \mathbb{R}^K$ be i.i.d. sub-Gaussian vectors with covariance $\Sigma \succ 0$. $L_T(\delta) = \ell(h_S(X_T) + \phi_T^\top \delta, y_T)$, where $\ell(\eta, y)$ is twice-differentiable and $\nabla_\eta^2 \ell(\eta, y) \geq \kappa > 0$ uniformly. We aim to show that, for sufficiently large $n_T$, with high probability over the sample,*

$$L_T(\mathbf{a}) - L_T(\mathbf{b}) - \langle \mathbf{a} - \mathbf{b}, \nabla L_T(\mathbf{b}) \rangle \geq c_1 n_T \|\mathbf{a} - \mathbf{b}\|_2^2 - c_2 \|\mathbf{a} - \mathbf{b}\|_1^2.$$

*for all $\mathbf{a}, \mathbf{b} \in \mathbb{R}^K$, where $c_1, c_2 > 0$ are constants depending on $\kappa$ and $\Sigma$.*

*Proof.* Define $\boldsymbol{h} = \mathbf{a} - \mathbf{b}$. By Taylor's theorem, there exists a point $\tilde{\delta}$ on the line segment between $\mathbf{a}$ and $\mathbf{b}$ such that

$$L_T(\mathbf{a}) - L_T(\mathbf{b}) - \langle \boldsymbol{h}, \nabla L_T(\mathbf{b}) \rangle = \frac{1}{2} \boldsymbol{h}^\top \nabla^2 L_T(\tilde{\delta}) \boldsymbol{h},$$

where

$$\nabla^2 L_T(\tilde{\delta}) = \sum_{i=1}^{n_T} \nabla_\eta^2 \ell(h_S(X_i) + \phi_i^\top \tilde{\delta}, y_i) \phi_i \phi_i^\top \succeq \kappa \sum_{i=1}^{n_T} \phi_i \phi_i^\top.$$

Defining the empirical covariance $\hat{\Sigma} = \frac{1}{n_T} \sum_{i=1}^{n_T} \phi_i \phi_i^\top$, it follows that

$$\boldsymbol{h}^\top \nabla^2 L_T(\tilde{\delta}) \boldsymbol{h} \geq \kappa n_T \boldsymbol{h}^\top \hat{\Sigma} \boldsymbol{h}.$$

Under the assumption that $\{\phi_i\}$ are i.i.d. sub-Gaussian with $\mathbb{E}[\phi_i \phi_i^\top] = \Sigma \succ 0$, standard concentration results (e.g., Theorem 9 of Rudelson & Vershynin (2010), or Theorem 3.1 in Raskutti et al. (2010)) show that for $n_T$ on the order of $\frac{a \log K}{\lambda_{\min}(\Sigma)}$, the empirical covariance $\hat{\Sigma}$ satisfies a restricted eigenvalue inequality with high probability

$$\boldsymbol{h}^\top \hat{\Sigma} \boldsymbol{h} \geq \gamma_{\min} \|\boldsymbol{h}\|_2^2 - \tau \frac{\log K}{n_T} \|\boldsymbol{h}\|_1^2.$$

where $\gamma_{\min} > 0$ and $\tau > 0$ are constants depending on $\Sigma$ and the sub-Gaussian norm of $\phi_i$. Combining this restricted eigenvalue (RE) bound with the lower Hessian bound above yields

$$\boldsymbol{h}^\top \nabla^2 L_T(\tilde{\delta}) \boldsymbol{h} \geq \kappa n_T \left( \gamma_{\min} \|\boldsymbol{h}\|_2^2 - \tau \frac{\log K}{n_T} \|\boldsymbol{h}\|_1^2 \right)$$

Substitute back into the Taylor expansion, there exist constants $c_1$ and $c_2$ such that

$$L_T(\mathbf{a}) - L_T(\mathbf{b}) - \langle \mathbf{a} - \mathbf{b}, \nabla L_T(\mathbf{b}) \rangle \geq \underbrace{\tfrac{1}{2} \kappa \gamma_{\min}}_{c_1} n_T \|\mathbf{a} - \mathbf{b}\|_2^2 - \underbrace{\tfrac{1}{2} \kappa \tau}_{c_2} \|\mathbf{a} - \mathbf{b}\|_1^2.$$

This is precisely the Restricted Strong Convexity (RSC) condition. □

# B  PROOF OF THEOREM 3.1: CONVERGENCE GUARANTEE FOR ESTIMATION ERROR UNDER RSC

*Proof.* Given the definition of $\hat{\delta}$, there exists a subgradient $\boldsymbol{z} \in \partial\|\hat{\delta}\|_1$ such that

$$\nabla L(\hat{\delta}) + \lambda\boldsymbol{z} = \mathbf{0}$$

To bound the norm of the parameter $\delta$, with standard Lasso analysis under Restricted Strong Convexity (RSC) Van de Geer (2008) (justified in Appendix A), we will use the RSC condition of $L$ that

$$L(\mathbf{a}) - L(\mathbf{b}) - \langle \mathbf{a} - \mathbf{b}, \nabla L(\mathbf{b}) \rangle \geq c_1 n_T \|\mathbf{a} - \mathbf{b}\|_2^2 - c_2 \|\mathbf{a} - \mathbf{b}\|_1^2$$

As $\hat{\delta}$ minimized the penalized loss,

$$L(\hat{\delta}) - L(\delta^*) + \lambda(\|\hat{\delta}\|_1 - \|\delta^*\|_1) \leq 0$$

By the RSC condition

$$L(\hat{\delta}) - L(\delta^*) - \langle \hat{\delta} - \delta^*, \nabla L(\delta^*) \rangle \geq c_1 n_T \|\hat{\delta} - \delta^*\|_2^2 - c_2 \|\hat{\delta} - \delta^*\|_1^2$$

Define $\boldsymbol{d} = \hat{\delta} - \delta^*$

$$c_1 n_T \|\boldsymbol{d}\|_2^2 - c_2 \|\boldsymbol{d}\|_1^2 + \langle \boldsymbol{d}, \nabla L(\delta^*) \rangle + \lambda(\|\hat{\delta}\|_1 - \|\delta^*\|_1) \leq 0$$

By Hölder's inequality

$$\|\langle \boldsymbol{d}, \nabla L(\delta^*) \rangle\|_1 \leq \|\boldsymbol{d}\|_1 \|\nabla L(\delta^*)\|_\infty$$

By triangle inequality

$$\|\hat{\delta}\|_1 - \|\delta^*\|_1 \geq -\|\boldsymbol{d}\|_1$$

Under the assumption that

$$\|\nabla L(\delta^*)\|_\infty \leq c_3 \lambda$$
$$|\langle \boldsymbol{d}, \nabla L(\delta^*) \rangle| \leq c_3 \lambda \|\boldsymbol{d}\|_1$$

From standard Lasso analysis, we often assume

$$\|\boldsymbol{d}_{A^c}\|_1 \leq \|\boldsymbol{d}_A\|_1, \|\boldsymbol{d}\|_1 \leq 2\|\boldsymbol{d}_A\|_1$$

By Cauchy-Schwarz inequality

$$\|\boldsymbol{d}_A\|_1 \leq \sqrt{a}\|\boldsymbol{d}_A\|_2 \leq \sqrt{a}\|\boldsymbol{d}\|_2, \|\boldsymbol{d}\|_1^2 \leq 4a\|\boldsymbol{d}\|_2^2$$

$$0 \geq c_1 n_T \|\boldsymbol{d}\|_2^2 - c_2 \|\boldsymbol{d}\|_1^2 + \langle \boldsymbol{d}, \nabla L(\delta^*) \rangle + \lambda(\|\hat{\delta}\|_1 - \|\delta^*\|_1)$$

$$\geq c_1 n_T \|\boldsymbol{d}\|_2^2 - c_2 \|\boldsymbol{d}\|_1^2 - c_3 \lambda \|\boldsymbol{d}\|_1 - \lambda\|\boldsymbol{d}\|_1$$

$$\geq c_1 n_T \|\boldsymbol{d}\|_2^2 - 4c_2 a\|\boldsymbol{d}\|_2^2 - 2(c_3 + 1)\sqrt{a}\lambda\|\boldsymbol{d}\|_2$$

$$\|\boldsymbol{d}\|_2 \leq \frac{2\sqrt{a}(c_3 + 1)\lambda}{c_1 n_T - 4c_2 a}$$

with the condition $\lambda \asymp \sqrt{\log K/n_T}$, we can rewrite $\|\boldsymbol{d}\|_2^2 \leq \frac{Ca \log K}{n_T}$, where $C > 0$ depends on $c_1, c_2, c_3$ but not on $n_T$, $a$, or $K$, reaching a similar result to Li et al. (2022; 2023) in sparse high dimensional regression based transfer learning. □

**Remark 1.** The RSC condition is ensured by the sub-Gaussian design and curvature of the loss (Lemma A.1), which are standard in high-dimensional statistics Raskutti et al. (2010).

**Remark 2.** Sub-Gaussian concentration gives $\|\nabla L(\delta^*)\|_\infty \leq c_3 \lambda$ with high probability Vershynin (2018).

**Remark 3.** The condition $\lambda \asymp \sqrt{\log K/n_T}$ ensures compatibility between regularization and noise, standard in $\ell_1$-penalized M-estimation Van de Geer (2008).

**Remark 4.** The offset $h_S$ does not affect RSC because it is fixed during the optimization over $\delta$, leaving the curvature of $\ell$ and design $\phi_i$ as the drivers of convexity.

## C  CONSTRUCTION AND SELECTION WITH KNOCKOFFS

Following Candes et al. (2018), a Model-X knockoff matrix $\tilde{X} = [\tilde{X}^{(1)}, \ldots, \tilde{X}^{(p)}] \in \mathbb{R}^{n \times p}$ of a matrix constructed by horizontally stacked random vectors $X = [X^{(1)}, \ldots, X^{(p)}] \in \mathbb{R}^{n \times p}$. The knockoffs is constructed such that, for any subset $A \subseteq [p]$,

$$\left( X, \tilde{X} \right)_{\text{swap}(A)} \overset{d}{=} \left( X, \tilde{X} \right).$$

where $X^{(j)}$ denotes the $j$th column of $X$, $\left( X, \tilde{X} \right)_{\text{swap}(A)}$ is obtained by swapping the column entries $X^{(j)}$ and $\tilde{X}^{(j)}$ for any $j \in A$. Crucially, $\tilde{X}$ must be constructed conditional on $X$ but independent of $y$ to ensure $\tilde{X} \perp\!\!\!\perp y \mid X$.

We set variable importance measure as coefficients: $Z_k = |\hat{\delta}'_k(\lambda)|$, $\tilde{Z}_k = |\hat{\delta}'_{k+K}(\lambda)|$. Alternatively, we can also use $Z_k = \sup\{\lambda \geq 0 : \hat{\delta}'_k(\lambda) \neq 0\}$, the lambda value where each feature/knockoff enters the lasso path (meaning becomes nonzero). The knockoff filter works by comparing the $Z_k$'s to the $\tilde{Z}_k$'s and selecting only variables that are clearly better than their knockoff copy. The reason why this can be done is that, by construction of the knockoffs, the null (not related to $y$) statistics are pairwise exchangeable. This means that swapping the $Z_k$ and $\tilde{Z}_k$'s corresponding to null variables leaves the koint distribution of $(Z_1, \ldots, Z_K, \tilde{Z}_1, \ldots, \tilde{Z}_K)$ unchanged. Once the $Z_k$ and $\tilde{Z}_k$'s have been computed, different contrast functions can be used to compare them. In general, we must choose an anti-symmetric function $a$ and we compute the symmetrized knockoff statistics $W_k = a(Z_k, \tilde{Z}_k) = -a(\tilde{Z}_k, Z_k)$ such that $W_k$ indicates that $X_k$ appears to be more important than its own knockoff copy. We use difference of absolute values of coefficients by default, but many other alternatives (like signed maximum) are also possible.

## D   PROOF OF THEOREM 3.2: STABILITY SELECTION CONTROL

*Proof.* **PFER Control:** For each null feature $k \in A^c$, the per-iteration selection probability satisfies $P(k \in \hat{A}^{[b]}) \leq \alpha$. This holds if $\tau$ is chosen to control the PFER at level $\alpha K$ in each iteration, following Meinshausen & Bühlmann (2010). Define $V_k^{[b]} = \mathbf{1}\{k \in \hat{A}^{[b]}\}$ as independent Bernoulli trials with $\mathbb{E}[V_k^{[b]}] \leq \alpha$. The selection frequency $\hat{\Pi}_k = \frac{1}{B}\sum_{b=1}^{B} V_k^{[b]}$ is a binomial proportion with $\mathbb{E}[\hat{\Pi}_k] \leq \alpha$. By Hoeffding's inequality:

$$P\left(\hat{\Pi}_k \geq \pi\right) \leq \exp\left(-2B(\pi - \alpha)^2\right) \quad \forall \pi > \alpha.$$

Summing over all null features and applying linearity of expectation:

$$\mathbb{E}\left[|\hat{A}(\pi) \cap A^c|\right] = \sum_{k \in A^c} P\left(\hat{\Pi}_k \geq \pi\right) \leq |A^c| \exp\left(-2B(\pi - \alpha)^2\right).$$

**FDR Control:** Let $V^{[b]} = |\hat{A}^{[b]} \cap A^c|$ and $R^{[b]} = |\hat{A}^{[b]}|$. By the knockoff filter guarantee from Theorem 3.1 in Candes et al. (2018), each $\tau$ ensures $\mathbb{E}\left[\frac{|\hat{A}^{[b]} \cap A^c|}{|\hat{A}^{[b]}| \vee 1}\right] = \mathbb{E}\left[\frac{V^{[b]}}{R^{[b]} \vee 1}\right] \leq q$. The stabilized FDR satisfies:

$$\mathrm{FDR}(\hat{A}(\pi)) = \mathbb{E}\left[\frac{|\hat{A}(\pi) \cap A^c|}{|\hat{A}(\pi)| \vee 1}\right] \leq \mathbb{E}\left[\frac{\sum_{b=1}^{B} V^{[b]}}{B\pi}\right] \quad \text{(since } \hat{\Pi}_k \geq \pi \implies \sum_{b=1}^{B} \mathbf{1}\{k \in \hat{A}^{[b]}\} \geq B\pi\text{)}$$

$$= \frac{1}{B\pi}\sum_{b=1}^{B} \mathbb{E}\left[\frac{V^{[b]}}{R^{[b]} \vee 1} \cdot R^{[b]}\right] = \frac{1}{B\pi}\sum_{b=1}^{B} \mathbb{E}\left[R^{[b]} \cdot \mathbb{E}\left[\frac{V^{[b]}}{R^{[b]} \vee 1} \mid R^{[b]}\right]\right]$$

$$\leq \frac{1}{B\pi}\sum_{b=1}^{B} \mathbb{E}\left[R^{[b]} \cdot q\right] = \frac{q}{B\pi}\sum_{b=1}^{B} \mathbb{E}\left[R^{[b]}\right].$$

where the last line is by Theorem 3.1 in Candes et al. (2018). From Proposition 1 and Appendix A.2 in Ren et al. (2023), the geometric thinning inequality $\sum_{b=1}^{B} \mathbb{E}[R^{[b]}] \geq \frac{\mathbb{E}[|\hat{A}(\pi)|]}{1-(1-\pi)^B}$ holds because each feature's selection events are independent across $B$ iterations. Substituting this bound

$$\mathrm{FDR}(\hat{A}(\pi)) \leq \frac{q}{B\pi}\sum_{b=1}^{B} \frac{\mathbb{E}[|\hat{A}(\pi)|]}{1-(1-\pi)^B} = \frac{q}{1-(1-\pi)^B}.$$

$\square$

# E FDR CONTROL OF NAIVE SGSHIFT

Given the assumption of i.i.d. observations and the exponential family distribution to generate the dependent variable $y$, $f(X_i)$ determines the $\mathbb{E}[y_i \mid X_i]$ under domain $S$, and $\delta$ captures the shift.

The negative log-likelihood function of $\delta$ can be written as

$$L(\delta) = \sum_{i=1}^{n_T} \left\{ \psi\big(f(X_i) + \phi_i^\top \delta\big) - y_i\big(f(X_i) + \phi_i^\top \delta\big) \right\} = \ell\big(f(X_T) + \phi_T^\top \delta, y_T\big)$$

where $\psi(\cdot)$ is uniquely determined by the link $g(\cdot)$, $\ell(\eta, y) = \psi(\eta) - y\eta$ where $\eta = f(X) + \phi^\top \delta$.

We regularize the GAM loss with an $\ell_1$-penalty

$$\hat{\delta}(\lambda) = \arg\min_{\delta \in \mathbb{R}^K} \left\{ \sum_{i=1}^{n_T} \big(\psi(\eta_i) - y_i \eta_i\big) + \lambda \|\delta\|_1 \right\}$$

The score vector is

$$\nabla L(\delta) = \sum_{i=1}^{n_T} \left[ \psi'\big(f(X_i) + \phi_i^\top \delta\big) - y_i \right] \phi_i$$

Evaluated at $\delta = 0$, $\gamma := \nabla L(\delta)\big|_{\delta=0} = \sum_{i=1}^{n_T} \left[ \psi'\big(f(X_i)\big) - y_i \right] \phi_i$, where $\psi'(\cdot)$ is the canonical mean function.

By the Karush–Kuhn–Tucker (KKT) conditions, $\hat{\delta}_j(\lambda) \neq 0$ iff $|\gamma_j| > \lambda$; hence selection of $j$ depends only on the distribution of $\gamma_j$.

Assume each row $\phi_i$ is sub-Gaussian with i.i.d. coordinates and that every coordinate of $\phi_i$ and $y_i$ has been centered and variance-normalized.

Let $\sigma_\gamma^2 := \mathbb{V}(y_i \mid X_i) = \psi''\big(f(X_i)\big)$, where $\psi''(\cdot)$ is the variance function of the canonical exponential-family model.

Let the true parameter be $\delta^* \in \mathbb{R}^K$ with support $A \subseteq [K]$, $|A| = a$, so $\delta_j^* = 0$ for $j \in A^c$.

**Null coordinates.** Let $j$ be a null coordinate $j \in A^c$ among $K - a$ null coordinates.

Because $\delta_j^* = 0$,

$$\mathbb{E}[y_i - \psi'(f(X_i))|X_i] = 0, \mathbb{E}\big[\big(y_i - \psi'(f(X_i))\big)\phi_{ij}\big] = \mathbb{E}\left[\mathbb{E}[y_i - \psi'(f(X_i))|X_i]\phi_{ij}\right] = 0$$

Define $Z_{ij} = \big(y_i - \psi'(f(X_i))\big)\phi_{ij}$, $\{Z_{ij}\}_{i=1}^{n_T}$ are i.i.d., mean-zero, and sub-Gaussian.

$$\mathbb{V}[Z_{ij}] = \mathbb{E}\big[\big(y_i - \psi'(f(X_i))\big)^2 \phi_{ij}^2\big] = \mathbb{E}\left[\psi''(f(X_i))\phi_{ij}^2\right] = \mathbb{E}\left[\psi''(f(X_i))\right] = \sigma_\gamma^2$$

where the second equality follows by the law of total variance and independence between $\phi_{ij}$ and $y_i \mid X_i$.

Under mild moment conditions, the Lindeberg-Feller central limit theorem (CLT) implies

$$\frac{1}{\sqrt{n}} \sum_i \left\{ \big(y_i - \psi'(f(X_i))\big)\phi_{ij} \right\} \xrightarrow{d} N\big(0, \sigma_\gamma^2\big) \quad \text{if } j \in A^c$$

Therefore, for null coordinates, we have

$$\frac{1}{\sqrt{n}}\gamma_j = -\frac{1}{\sqrt{n}} \sum_i \big(y_i - \psi'(f(X_i))\big)\phi_{ij} \xrightarrow{d} N\big(0, \sigma_\gamma^2\big) \quad \text{if } j \in A^c$$

**False-selection probability and plug-in mFDR estimate.** Because $\hat{\delta}_j(\lambda) \neq 0$ iff $|\gamma_j| > \lambda$, the null-coordinate error rate is

$$Pr(j \text{ selected}|j \in A^c) = Pr(|\gamma_j| > \lambda) = 2\Big\{1 - \Phi\big(\frac{\lambda}{\sqrt{n}\sigma_\gamma}\big)\Big\}$$

where $\Phi$ is the standard normal CDF.

Following Miller & Breheny (2019), the marginal FDR is

$$\text{mFDR}(\lambda) = \frac{\mathbb{E}[\#\text{False Discoveries}]}{\mathbb{E}[\#\text{Selected}]}$$

Plugging in the null probability above yields

$$\widehat{\text{FDR}}(\lambda) = \min\Big\{\frac{2(K-a)\big(1 - \Phi\big(\lambda/\sqrt{n}\sigma_\gamma\big)\big)}{|\hat{A}(\lambda)| \vee 1}, 1\Big\}$$

A practical one-pass rule that controls FDR at level $\alpha$ is

$$\hat{\lambda}_\alpha = \min\Big\{\lambda : \widehat{\text{FDR}}(\lambda) \leq \alpha\Big\}$$

## F DATASETS

All datasets are listed as below, and the full preprocessing code from raw data, together with the preprocessed data, are available in the source code, except restricted access COVID-19 Hospitalization data where we provide detailed fetching code and data version information from NIH All of Us Research Program of Us Research Program Investigators (2019). Standardization is performed within the pipeline to ensure that features with larger values don't disproportionately influence the $\ell_1$ regularization penalty.

|                | Diabetes readmission      | COVID-19    | SUPPORT2          |
|----------------|---------------------------|-------------|-------------------|
| Total samples  | 73,615                    | 16,187      | 9,105             |
| Features       | 33                        | 30          | 64                |
| Source size    | 49,213                    | 11,268      | 5,453             |
| Target size    | 24,402                    | 2,219       | 1,817             |
| Domain split   | Emergency room admission  | New variant | Death in hospital |

Table 2: Dataset summary.

**Diabetes 30-Day Readmission**    The Diabetes 130-US Hospitals dataset, available through the UCI Machine Learning Repository (`https://archive.ics.uci.edu/dataset/296/diabetes+130-us+hospitals+for+years+1999-2008`), comprises 101,766 encounters of diabetic patients across 130 U.S. hospitals between 1999-2008 Strack et al. (2014). We fetch the data following TableShift's procedure Gardner et al. (2023). We define the source domain as 49,213 non-ER admissions (elective or urgent) with 25,196 readmitted patients, and the target domain as 24,402 ER admissions with 10,684 readmitted patients, with the binary classification task being prediction of 30-day readmission risk.

**COVID-19 Hospitalization**    The COVID-19 cohort is part of the NIH All of Us Research Program of Us Research Program Investigators (2019), a (restricted access) dataset containing electronic health records for 16,187 patients diagnosed with COVID-19 between 2020-2022. Features include demographic variables (age, gender, race), temporal indicators (diagnosis date relative to Omicron variant emergence), comorbidity status for 13 chronic conditions (diabetes, COPD), and diagnostic context (EHR vs. claims-based). We partition the data into three temporal groups: a source domain of 11,268 patients diagnosed prior to the beginning of 2022 with 2,541 patients hospitalized, a target domain of 2,219 patients diagnosed in January 2022 (early Omicron era) with 359 patients hospitalized. The binary classification task predicts hospitalization status (inpatient vs. outpatient).

**SUPPORT2 Hospital Charges**    From the Study to Understand Prognoses Preferences Outcomes and Risks of Treatment (SUPPORT2), publicly available via the UCI repository (`https://archive.ics.uci.edu/dataset/880/support2`) containing 9,105 critically ill patients Connors et al. (1995). The source domain is specified as 5,453 patients who survived hospitalization and the target domain as 1,817 in-hospital deaths. The regression task is defined as a prediction of $\log_{10}$(total hospital costs per patient).

## G MODEL HYPERPARAMETERS

We used standard implementations of classical machine learning models from `scikit-learn`, with hyperparameters either set to commonly used defaults or manually tuned for stability and performance. Supplementary table 3 summarizes the key hyperparameters for each model. Unless otherwise stated, all models were trained using their default solver settings. Random seeds were fixed via `random_state` to ensure reproducibility.

| Model | Hyperparameters |
|---|---|
| Decision Tree (Classifier) | `max_depth=4, random_state={seed}` |
| Support Vector Machine (Classifier) | `kernel='rbf', C=1.0, probability=True, random_state={seed}` |
| Gradient Boosting Classifier | `n_estimators=100, random_state={seed}` |
| Logistic Regression | `max_iter=200, random_state={seed}` |
| Decision Tree (Regressor) | `max_depth=4, random_state={seed}` |
| Support Vector Machine (Regressor) | `kernel='rbf', C=1.0` |
| Linear Regression | default settings |
| Gradient Boosting Regressor | `n_estimators=100, random_state={seed}` |

Table 3: Hyperparameters used for each model. The same random seed (`{seed}`) was applied across models where applicable to ensure consistency.

# H    ABLATIONS

| | Sparse simulations | | | Dense simulations | | |
|---|---|---|---|---|---|---|
| Model Match | SGShift | SGShift-A | SGShift-K | SGShift | SGShift-A | SGShift-K |
| **Diabetes Readmission** | | | | | | |
| Matched | 0.80 | 0.81 | **0.90** | 0.71 | 0.72 | **0.85** |
| Mismatched | 0.79 | 0.81 | **0.86** | 0.72 | 0.71 | **0.78** |
| **COVID-19** | | | | | | |
| Matched | 0.86 | 0.88 | **0.99** | 0.80 | 0.83 | **0.93** |
| Mismatched | 0.85 | 0.80 | **0.97** | 0.76 | 0.76 | **0.91** |
| **SUPPORT2** | | | | | | |
| Matched | 0.92 | 0.94 | **0.96** | 0.89 | 0.89 | **0.92** |
| Mismatched | 0.86 | 0.88 | **0.95** | 0.88 | 0.89 | **0.92** |

Table 4: **Performance (AUC) of SGShift variants in identifying shifted features.** AUC of SGShift, SGShift-A, and SGShift-K in sparse (left) and dense (right) semi-synthetic simulations. Matched refers to when generator and base model are the same, mismatched when they differ.

# I SAMPLE SIZE

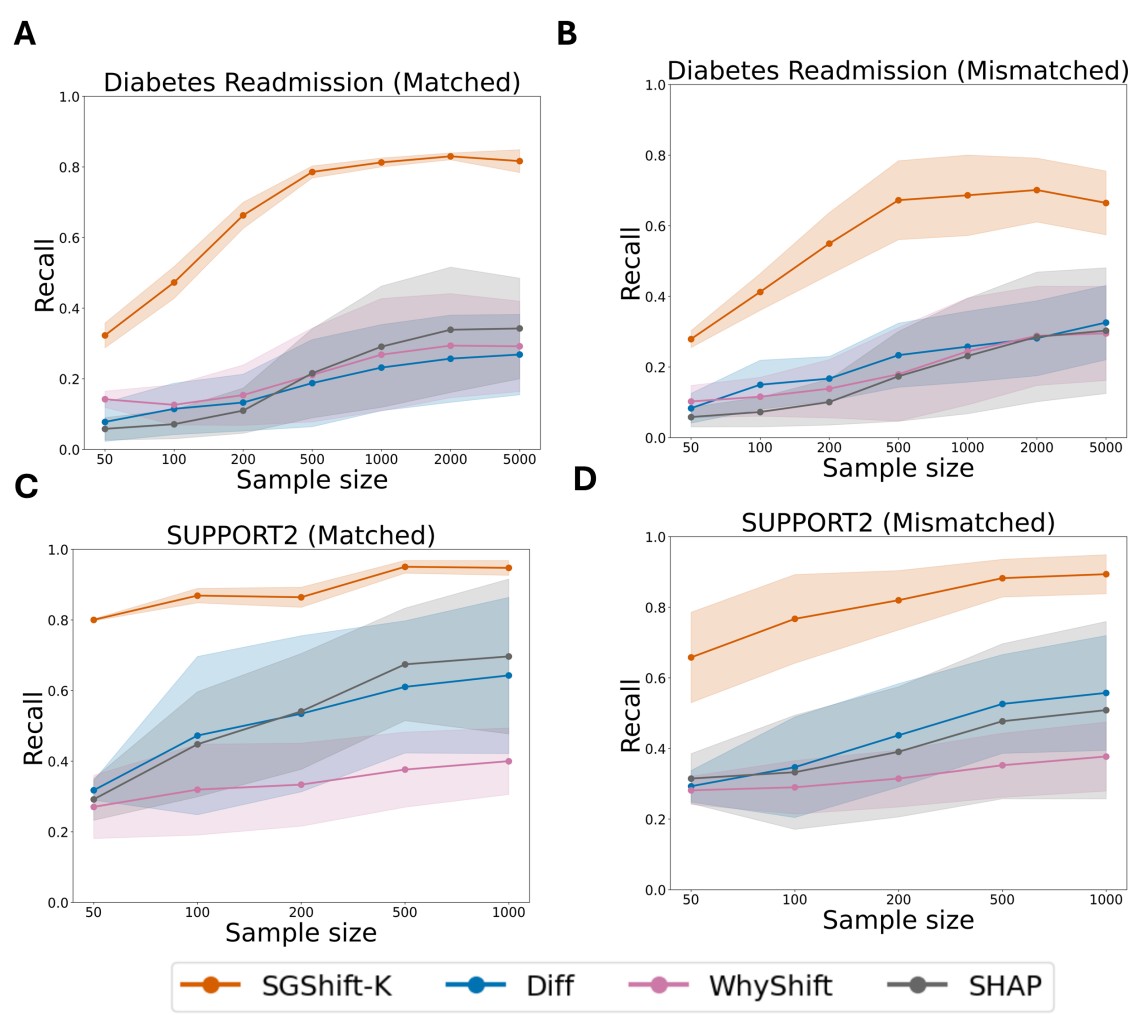

Figure 4: **Performance across sample sizes.** Sample size is varied from 50 to 1000. 95% CI's are shown across 16 simulation settings. Recall is measured at fixed FPR 5%.

## J  GLOBAL CALIBRATION SHIFTS

**Global shifts.** It may be true that all features are shifted in the same direction in a new domain due to differences in sensor calibration. In this case identifying specific shifted features may be difficult as all are perturbed slightly. We simulate this global effect where only a few true features having a conditional shift, and the rest are perturbed by noise with absolute values from 0.01 to 0.3 while the absolute values of true shifts are 3. Results are reported in Figure 5. SGShift-K still strongly identifies true shifted features with AUC > 0.9, even when all features are shifted slightly, and individual features are not over or under prioritized. Interestingly, for all methods, performance is relatively unchanged as the scale of the background shift increases. This may be due to the intercept term accumulating the background shift, as opposed to attributing it to any individual feature.

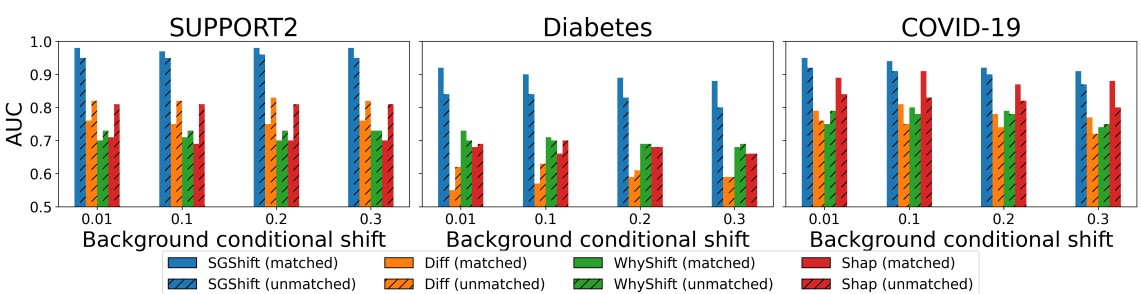

Figure 5: **Global calibration shift performance.** Performance as a background conditional shift is increased in scale. X-axis represents strength of the background shift, as 0.01x-0.3x the true shift magnitude.

## K INTERACTION SHIFTS

**Interaction shifts.** We assess each method's performance in identifying features which shift in the interaction space in the SUPPORT2 dataset, where features are continuous. The goal is to detect individual features contributing to shift through interactions with other features. We consider two cases of SGShift-K, underspecified, where the basis function does not include interaction terms, and overspecified, where SGShift contains both second and third tier interactions. Results are reported in Table 5. In both cases, regardless of how the basis function is specified, SGShift displays strong performance.

| | Diff | WhyShift | SHAP | SGShift-K – underspecified | SGShift-K – overspecified |
|---|---|---|---|---|---|
| Matched | 0.75 | 0.73 | 0.80 | **0.92** | 0.90 |
| Mismatched | 0.72 | 0.73 | 0.78 | **0.89** | 0.87 |

Table 5: **Performance in detecting interaction shifts.**

## L    HIGH DIMENSIONAL SHIFTS

**High dimensional shifts.** We simulate high dimensional data with 1000 samples and 500, 200, and 100 features, for each of the 16 model configurations. In each case, 20% of the features are shifted. Results are reported below. SGShift-K maintains strong performance (all AUC > 0.89) even as the number of features is half the number of samples where other methods lose performance.

| Model Match | Diff | WhyShift | SHAP | SGShift-K |
|---|---|---|---|---|
| **500 Features** | | | | |
| Matched | 0.57 | 0.50 | 0.61 | **0.92** |
| Unmatched | 0.57 | 0.51 | 0.62 | **0.89** |
| **200 Features** | | | | |
| Matched | 0.87 | 0.53 | 0.88 | **0.99** |
| Unmatched | 0.86 | 0.53 | 0.86 | **0.97** |
| **100 Features** | | | | |
| Matched | 0.93 | 0.57 | 0.93 | **1.00** |
| Unmatched | 0.94 | 0.57 | 0.92 | **1.00** |

Table 6: **Performance in identifying shifted features across feature dimensionalities.** AUC of Diff, WhyShift, SHAP, and SGShift-K for different numbers of features.

## M  CORRELATED FEATURES

**Correlated features** We conduct an experiment by simulating 1000 samples in each domain and 500 features (100 shifted), and varying the maximum feature correlation $\rho$ from 0.1 to 0.9, with i-th and j-th predictors correlated as $\rho^{|i-j|}$. Results are reported in the figure below. In the presence of shifted feature correlation, SGShift-K is still able to strongly identify shifted features, likely due to knockoff's innate ability at handling feature correlations.

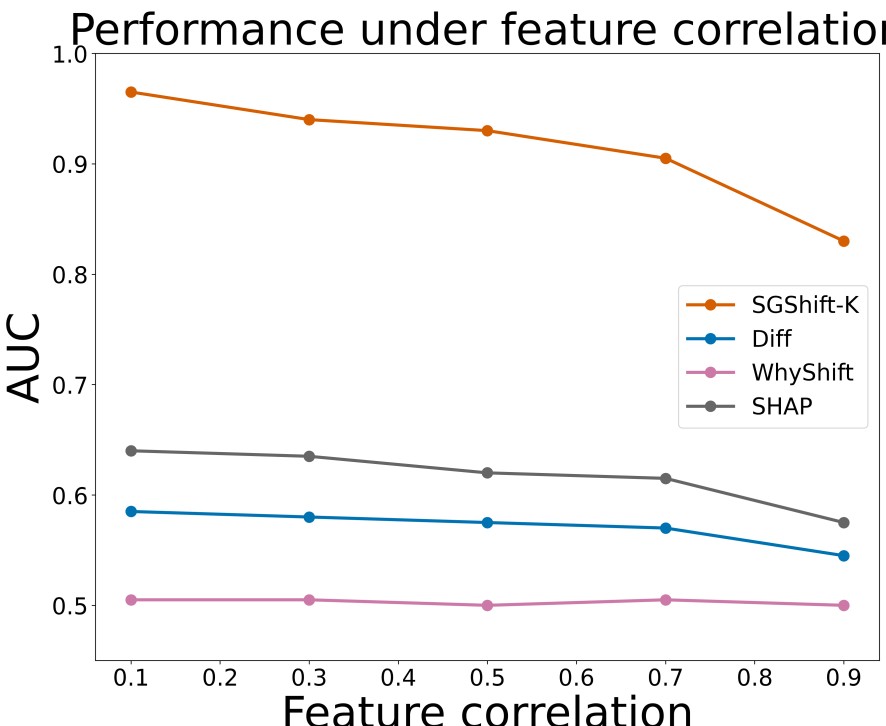

Figure 6: **Feature correlation performance.** Ability to identify shifted features as features become more correlated.

# N DIAGNOSTICS

**Diagnostics** To understand the level of concept shift in real data, we add an experiment testing how much performance can be recovered with the inverse propensity weighting procedure from WhyShift to account for covariate shift. Results are reported below. At most this can recover 15% of the difference, and in many cases it actually reduces performance, likely due to its reliance on sufficient target domain sample size, which as in the Covid-19 example is small.

| Dataset | Decision Tree | Gradient Boosting | Regression | SVM |
|---------|---------------|-------------------|------------|-----|
| Support2 | -3.08% | 11.81% | -6.78% | 2.39% |
| Diabetes | -11.50% | 2.64% | -0.79% | 14.30% |
| Covid-19 | -79.09% | -21.11% | -7.38% | -4.90% |

Table 7: **Relative performance (%) after correcting for covariate shift with IPW.**

## O    SPARSITY IN REAL DATA

**Sparsity in real data.** We perform additional performance recovery experiments on datasets with known concept shift from WhyShift's datasets. We train models across 31 state-state pairs in the ACS income datasets, and in 87 cases of concept shift, 78 of these can have model performance completely recovered by less than 1/3 of the total features, 42 of which require less than 10% of features. The remaining 9 can be recovered with less than 50% of features.

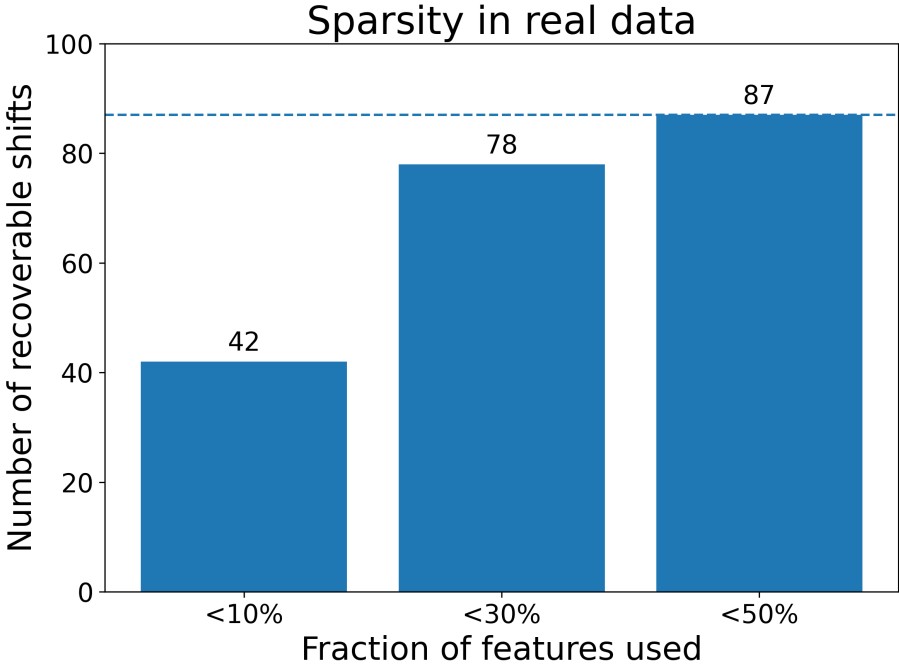

Figure 7: **Features needed to correct for concept shift in real data.**.

## P    SIGNAL TO NOISE EFFECT

**Varying signal to noise.** We simulate signal-to-noise ratios from 1 to 16 by simulating 1000 samples in each domain with 200 features, 40 of which induce concept shift, maximum feature correlation 0.7 with i-th and j-th predictors correlated as $\rho^{|i-j|}$. We vary the noise variance of the induced concept shift so the signal-to-noise ratio is 1 to 16. Results are reported below. We additionally include naive SGShift and SGShift-A in the SNR study because knockoff-based methods rely on accurate estimation of the feature covariance structure; when the induced concept shift becomes extremely noisy, this estimation becomes less stable, which can reduce knockoff power. Even so, across all SNR regimes, all SGShift variants substantially outperform baseline methods.

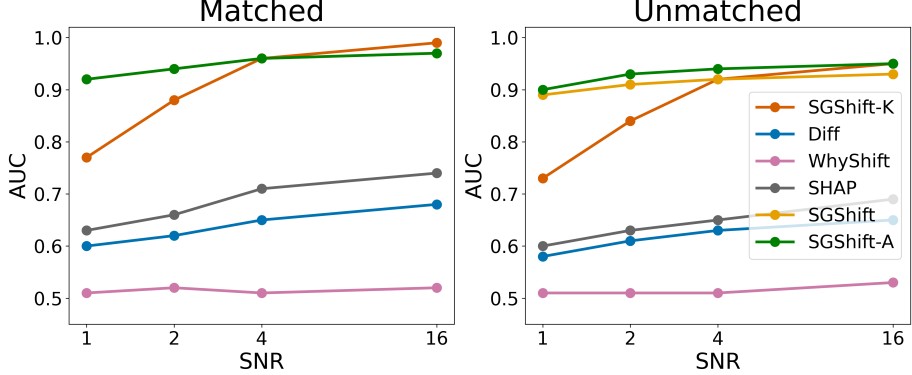

Figure 8: **Signal to noise ratio.** Ability to identify shifted features as signal to noise ratio changes.