# OpenReview forum: "Explaining Concept Shift with Interpretable Feature Attribution"
_ICLR.cc/2026/Conference — Submitted to ICLR 2026_

### Official Review · Reviewer_BPYY · 2025-10-25

**Soundness:** 3
**Presentation:** 3
**Contribution:** 2
**Rating:** 4
**Confidence:** 3

**Summary:**

This paper proposes SGShift, a model for detecting concept shift in tabular data by attributing performance difference to a sparse set of shifted features within the feature selection framework. Three variants of the method are introduced: SGShift, SGShift-A, and SGShift-K, with the latter two addressing source model mismatch and controlling the false discovery rate. The model is evaluated on three medical datasets.

**Strengths:**

The paper is clearly written and well structured, and the three variants of the proposed method are clearly organized.

**Weaknesses:**

1. Table 1 and the loss curve in Figure 2 is missing confidence intervals.

2. The experimental setup seems somewhat limited. How are the features in each dataset selected for inducing the concept shift? How does the model performance change when selecting high-correlation features versus low-correlation features? How do the results vary under different signal-to-noise ratios? I believe a more detailed analysis in these aspects would strengthen the paper.

3. How do SGShift, SGShift-A, and SGShift-K perform in each scenario? I think including such ablation studies could provide deeper insights into the contributions of each variant.

4. I don’t fully understand part B in Figure 2. How exactly are features added during the SGShift updates? Additionally, many other methods have been proposed to address concept shift, how would their loss curves compare when applied on top of SGShift?

**Questions:**

Please refer to the weakness section above.

---

> ### Author Response · Authors · 2025-11-21
> **Response to Reviewer BPYY (1/2)**
>
> Thank you for the thoughtful summary of our work and for highlighting the clarity in our presentation. We summarize each review comment and provide a point-by-point response below. The manuscript has been revised accordingly, incorporating 6 new analyses and 1 update to the main table, and 6 new supplementary figures/tables. This includes 3 new simulation settings, ablation studies, sparsity validation, and covariate shift analysis.
>
> > Table 1 and the loss curve in Figure 2 is missing confidence intervals.
>
> Thank you for the comment. We have added 95% CI's aggregated across model configurations to Table 1 as per your suggestion. The loss curve in Figure 2 does not have variation, as in it we are showing our (deterministic) method on one dataset.
>
> > The experimental setup seems somewhat limited. How are the features in each dataset selected for inducing the concept shift? How does the model performance change when selecting high-correlation features versus low-correlation features? How do the results vary under different signal-to-noise ratios? I believe a more detailed analysis in these aspects would strengthen the paper.
>
> Thank you for the comment. The reviewer raises 3 questions, which we answer in turn.
>
> 1. We select features in each dataset randomly. We've updated the experimental setup section to reflect this at line 284.
> 2. For performance under strong correlation, we conduct an experiment by simulating 1000 samples in each domain and 500 features (with 100 randomly selected features inducing concept shift), and varying the maximum feature correlation $\rho$ from 0.1 to 0.9, with i-th and j-th predictors correlated as $\rho^{|i−j|}$. Results are reported below and in Appendix M. Even under strong feature correlation, SGShift-K continues to identify shifted features accurately. This robustness arises because the knockoff framework explicitly constructs control features that mirror the dependence structure among predictors.
>
> Correlation 0.9
> | Group     | Diff | WhyShift | SHAP | SGShift-K |
> |----------|------|----------|------|-----------|
> | Matched   | 0.54 | 0.50     | 0.56 | 0.85      |
> | Unmatched | 0.55 | 0.50     | 0.59 | 0.81      |
>
> Correlation 0.5
> | Group     | Diff | WhyShift | SHAP | SGShift-K |
> |----------|------|----------|------|-----------|
> | Matched   | 0.57 | 0.50     | 0.62 | 0.95      |
> | Unmatched | 0.58 | 0.50     | 0.62 | 0.91      |
>
> Correlation 0.1
> | Group     | Diff | WhyShift | SHAP | SGShift-K |
> |----------|------|----------|------|-----------|
> | Matched   | 0.58 | 0.51     | 0.64 | 0.98      |
> | Unmatched | 0.59 | 0.50     | 0.64 | 0.95      |
>
> 3. Signal to noise. We simulate signal-to-noise ratios from 1 to 16 by simulating 1000 samples in each domain with 200 features, 40 of which induce concept shift, maximum feature correlation 0.7 with i-th and j-th predictors correlated as $\rho^{|i−j|}$. We vary the noise variance of the induced concept shift so the signal-to-noise ratio is 1 to 16. Results are reported below and added to Appendix P. We additionally include naive SGShift and SGShift-A in the SNR study because knockoff-based methods rely on accurate estimation of the feature covariance structure; when the induced concept shift becomes extremely noisy, this estimation becomes less stable, which can reduce knockoff power. Even so, across all SNR regimes, all SGShift variants substantially outperform baseline methods.
>
>
> | SNR (Matched)    | Diff | WhyShift | SHAP | SGShift | SGShift-A | SGShift-K |
> | ------ | ---- | -------- | ---- | ------- | --------- | --------- |
> | **16** | 0.68 | 0.52     | 0.74 | 0.97    | 0.97      | 0.99      |
> | **4**  | 0.65 | 0.51     | 0.71 | 0.96    | 0.96      | 0.96      |
> | **2**  | 0.62 | 0.52     | 0.66 | 0.94    | 0.94      | 0.88      |
> | **1**  | 0.60 | 0.51     | 0.63 | 0.92    | 0.92      | 0.77      |
>
>
> | SNR (Unmatched)    | Diff | WhyShift | SHAP | SGShift | SGShift-A | SGShift-K |
> | ------ | ---- | -------- | ---- | ------- | --------- | --------- |
> | **16** | 0.65 | 0.53     | 0.69 | 0.93    | 0.95      | 0.95      |
> | **4**  | 0.63 | 0.51     | 0.65 | 0.92    | 0.94      | 0.92      |
> | **2**  | 0.61 | 0.51     | 0.63 | 0.91    | 0.93      | 0.84      |
> | **1**  | 0.58 | 0.51     | 0.60 | 0.89    | 0.90      | 0.73      |
>
>
> Together with reviewer yhDC's suggestions, our work now features 8 total simulation settings (sparse, dense, sample size, global, interaction, high dimensional, highly correlated, and noisy shifts), which we believe constitutes very comprehensive experimental coverage.

---

> ### Author Response · Authors · 2025-11-21
> **Response to Reviewer BPYY (2/2)**
>
> > How do SGShift, SGShift-A, and SGShift-K perform in each scenario? I think including such ablation studies could provide deeper insights into the contributions of each variant.
>
> Thank you for the comment, we will provide clarification. In the feature selection tasks, we intend SGShift-K to be the primary tool, and when assessing recovery of model performance, we use naive SGShift or SGShift-A if the model is misfit. We have made this more clear in the main text and figure captions by explicitly mentioning which one is used for each task.
>
> Additionally, we have added ablation studies of feature selection to Appendix H. See results below. We see SGShift-K is indeed the most powerful in identifying shifted features.
>
> Sparse:
> | Dataset              | Group     | SGShift | SGShift-A | SGShift-K |
> | -------------------- | --------- | ------- | --------- | --------- |
> | SUPPORT2             | Matched   | 0.92    | 0.94      | 0.96      |
> | SUPPORT2             | Unmatched | 0.86    | 0.88      | 0.95      |
> | Diabetes Readmission | Matched   | 0.80    | 0.81      | 0.90      |
> | Diabetes Readmission | Unmatched | 0.79    | 0.81      | 0.86      |
> | COVID-19             | Matched   | 0.86    | 0.88      | 0.99      |
> | COVID-19             | Unmatched | 0.85    | 0.8       | 0.97      |
>
> Dense:
> | Dataset              | Group     | SGShift | SGShift-A | SGShift-K |
> | -------------------- | --------- | ------- | --------- | --------- |
> | SUPPORT2             | Matched   | 0.89    | 0.89      | 0.91      |
> | SUPPORT2             | Unmatched | 0.88    | 0.89      | 0.92      |
> | Diabetes Readmission | Matched   | 0.71    | 0.72      | 0.85      |
> | Diabetes Readmission | Unmatched | 0.72    | 0.71      | 0.78      |
> | COVID-19             | Matched   | 0.80    | 0.83      | 0.93      |
> | COVID-19             | Unmatched | 0.76    | 0.76      | 0.91      |
>
> > I don't fully understand part B in Figure 2. How exactly are features added during the SGShift updates? Additionally, many other methods have been proposed to address concept shift, how would their loss curves compare when applied on top of SGShift?
>
> Thank you for the comment. The reviewer raises two points, which we answer in turn.
>
> 1. Features are added in this plot by varying the penalty parameter, such that more features will have nonzero coefficients. We updated the Figure caption to make this more clear.
> 2. Our goal is to identify the set of shifted features, not maximize model performance on the target distribution. Figure 2 is intended to serve as validation that concept shift can indeed be sparse. That is, the important feature of the plot is that performance has a sharp U shape (indicating sparsity of the difference), not whether the curve could be made still sharper by another method. We acknowledge that existing work does aim to address the challenge of performance drop under concept shift. Using SGShift's learned features to maximize model performance is an interesting future direction but we don't make any claims of improvement at that task in this paper.
>
> We believe that we have successfully addressed all of your major comments.  If you feel we have adequately addressed your concerns, we would be grateful if you would you consider updating your evaluation. Thank you again for your time spent reviewing and please let us know if you have any further questions.

---

### Official Review · Reviewer_spce · 2025-11-01

**Soundness:** 3
**Presentation:** 4
**Contribution:** 3
**Rating:** 6
**Confidence:** 2

**Summary:**

The paper proposes SGShift, a method to identify features responsible for concept shift by learning a sparse correction on top of a fixed source model rather than retraining two models. It combines a generalized additive model with l1 regularization and optional knockoff-based FDR control. Experiments include semi-synthetic setups where conditional shifts are simulated, and two real health case studies (COVID-19 and lupus). Results show SGShift detects shifted features more accurately than prior baselines in controlled conditions.

**Strengths:**

The method is well-motivated and theoretically grounded. Modeling conditional changes as a sparse correction to a fixed predictor is a reasonable and efficient approach. Semi-synthetic results are clear and consistent, showing reliable feature recovery under designed concept shifts. The theoretical analysis is sound, and the paper is clearly written and organized. The real-world examples are interesting, though they mostly serve as demonstrations rather than solid proof of conditional shift

**Weaknesses:**

1. Real-world concept shift validity is unclear
It is unclear whether the real-world studies truly exhibit concept shift. For COVID-19, a shift in clinical patterns is plausible, but many other factors (e.g. testing policy, vaccination, care protocols) also changed, affecting p(X) and label definitions. Without reweighting or calibration analyses to separate covariate or label shift, the claim of conditional change remains suggestive rather than proven. The same issue applies to the Diabetes and SUPPORT2 splits, where differences may come from case mix or selection, not true conditional changes.

2. Lupus case likely reflects representation shift.
The lupus cross-ancestry experiment attributes predictive differences to conditional changes, but across ancestries, measured features often differ in distribution and linkage. Unless features are harmonized or mapped to a shared representation, this is likely representation or covariate shift, not p(y∣X) change. The findings are still biologically interesting but should be framed accordingly.

3. No diagnostics separating shift types.
There lacks standard diagnostics such as source-to-target reweighting tests, conditional calibration curves, or reliability plots to confirm residual error after matching p(X). Without these, it’s hard to verify the central claim that SGShift recovers features driving concept shift rather than other forms of drift.

**Questions:**

1. For the COVID-19 study, did you check whether performance gaps remain after reweighting the source cohort to match the target covariate distribution?
2. How do you distinguish p(y∣X) changes from label or measurement drift, especially when care policies evolved over time?
3. In the lupus study, can you harmonize gene features or use pathway-level embeddings to ensure observed shifts are not due to ancestry-specific feature distributions?
4. Could you include simple diagnostics (e.g. reweighting or calibration plots) in the appendix to directly test for conditional vs covariate shift?

---

> ### Author Response · Authors · 2025-11-21
> **Response to Reviewer spce (1/2)**
>
> Thank you for the accurate summary of our method and for highlighting its theoretical soundness and reliable performance. We summarize each review comment and provide a point-by-point response below. The manuscript has been revised accordingly, incorporating 6 new analyses and 1 update to the main table, and 6 new supplementary figures/tables. This includes 3 new simulation settings, ablation studies, sparsity validation, and covariate shift analysis.
>
> > Real-world concept shift validity is unclear It is unclear whether the real-world studies truly exhibit concept shift. For COVID-19, a shift in clinical patterns is plausible, but many other factors (e.g. testing policy, vaccination, care protocols) also changed, affecting p(X) and label definitions. Without reweighting or calibration analyses to separate covariate or label shift, the claim of conditional change remains suggestive rather than proven. The same issue applies to the Diabetes and SUPPORT2 splits, where differences may come from case mix or selection, not true conditional changes.
>
> Thank you for the comment. We agree that real-world domain differences may reflect a mixture of shift types. To understand the level of concept shift in real data, we add an experiment testing how much performance can be recovered with the inverse propensity weighting procedure from https://github.com/namkoong-lab/whyshift [1] to account for covariate shift. Results are reported below and added to Appendix N. At most this can recover ~15% of the difference, and in many cases it actually reduces performance, likely due to the instability of inverse propensity weights when the target domain is small and the domains are easily separable (yielding propensity scores concentrated near 0 and 1 and a very small effective sample size).
>
> [1] Liu, Jiashuo, et al. "On the need for a language describing distribution shifts: Illustrations on tabular datasets." Advances in Neural Information Processing Systems 36 (2023): 51371-51408.
>
> | Dataset   | Decision Tree | Gradient Boosting | Regression | SVM    |
> |-----------|---------------|-------------------|-----------|--------|
> | Support2  | -3.08%        | 11.81%            | -6.78%    | 2.39%  |
> | Diabetes  | -11.50%       | 2.64%             | -0.79%    | 14.30% |
> | Covid-19  | -79.09%       | -21.11%           | -7.38%    | -4.90% |
>
> > Lupus case likely reflects representation shift. The lupus cross-ancestry experiment attributes predictive differences to conditional changes, but across ancestries, measured features often differ in distribution and linkage. Unless features are harmonized or mapped to a shared representation, this is likely representation or covariate shift, not p(y∣X) change. The findings are still biologically interesting but should be framed accordingly.
>
> Thank you for the comment. We agree different ancestries will have different background gene expression distributions. To test the effect of covariate shift, we project out the ancestry associated expression, train a model on Europeans and apply it to Asians. Without correction, the accuracy is 0.84, and with this correction, the accuracy is 0.82, indicating this ancestry correction did not significantly recover the performance drop. We do however agree we can't completely rule out other forms of shift contributing due to the complex nature of biological data and we acknowledge that the predicted shifted genes in this experiment are computational predictions in one dataset and significant further work would be needed to validate this. We've added discussion on lines 421-422 to put this finding in context.
>
> "although we acknowledge these findings would need further validation to ensure results are robust to the many hidden confounders present in biological data."
>
> > No diagnostics separating shift types. There lacks standard diagnostics such as source-to-target reweighting tests, conditional calibration curves, or reliability plots to confirm residual error after matching p(X). Without these, it's hard to verify the central claim that SGShift recovers features driving concept shift rather than other forms of drift.
>
> See comment above about diagnostics.
>
> > For the COVID-19 study, did you check whether performance gaps remain after reweighting the source cohort to match the target covariate distribution?
>
> See comment above about diagnostics.

---

> ### Author Response · Authors · 2025-11-21
> **Response to Reviewer spce (2/2)**
>
> > How do you distinguish p(y∣X) changes from label or measurement drift, especially when care policies evolved over time?
>
> Thank you for the comment. The reviewer raises 2 related questions.
> 1. Label shift is not necessarily distinct from changes in $p(y|X)$, as $p(y)$ can be modeled as a combination of $p(X)$ and $p(y|X)$. We acknowledge this is an important future direction to consider.
> 2. Measurement drift is also not necessarily distinct from $p(y|X)$, its one potential reason for these shifts. Our goal and data would not allow us to distinguish this from concept shift, but it is a downstream question that identifying shifted features may provide information into investigating. It would likely require collecting additional data to fully distinguish, and we acknowledge that this is an important future direction.
>
> We've added lines 447-450 to the discussion section about label shift and measurement shift.
>
> "Future work could include optimizing model performance by explicitly modeling the difference between datasets given the identified shifted features, disentangling various contributors to concept shift such as label or measurement drift, or extending SGShift to non-tabular data, e.g., images or graphs."
>
> > In the lupus study, can you harmonize gene features or use pathway-level embeddings to ensure observed shifts are not due to ancestry-specific feature distributions?
>
> See comment above about Lupus.
>
> > Could you include simple diagnostics (e.g. reweighting or calibration plots) in the appendix to directly test for conditional vs covariate shift?
>
> See comment above about diagnostics.
>
> We believe that we have strongly addressed your 3 major comments. If you feel we have adequately addressed your concerns, we would be grateful if you would you consider updating your evaluation. Thank you again for your time spent reviewing and please let us know if you have any further questions.

---

### Official Review · Reviewer_yhDc · 2025-11-04

**Soundness:** 3
**Presentation:** 3
**Contribution:** 3
**Rating:** 4
**Confidence:** 4

**Summary:**

The paper addresses the problem of detecting distribution shift, focusing on changes in the conditional distribution of labels given the input features and the labels $p(y|x)$. Under the common assumption that the distribution shifts are sparse, the authors frame the detection of shift between the source and target domains as a sparse regression problem over the input features. They propose SGShift, a method that learns a sparse correction to a predictor trained on the source domain and adapts it to the target distribution to identify which features are responsible for the shift. The paper also introduces variants that account for possible errors in the source model (SGShift-A) and control false feature discoveries using knockoffs (SGShift-K). The approach is evaluated on three healthcare datasets, Diabetes Readmission, COVID-19 Hospitalizations, and Support2, covering both semi-synthetic (simulated distribution shift) and real-world shifts, and the results show that SGShift often outperforms the baselines.

**Strengths:**

- The proposed approach and its variants are rigorous, supported by both theoretical analysis and thorough empirical validation. The formulation of distribution shift detection as a sparse regression problem is well-motivated, and the inclusion of variants that account for model misspecification and false discovery control adds robustness and credibility to the framework.

- While the idea of modeling distribution shift through sparsity assumptions is not entirely novel, and is widely used in causal representation  and invariant learning literature , the particular formulation of learning a sparse correction to the predictive model across domains is novel best of my knowledge.

- The paper is also well-presented overall. The methodology is easy to follow and the theoretical results are explained with sufficient intuition.

**Weaknesses:**

- The authors should cite relevant work on causal representation learning to properly justify the sparsity assumption. This assumption is well established in the causality literature through the concept of sparse mechanism shift, which posits that under the correct causal factorization, only a small subset of mechanisms is expected to vary across domains (Schölkopf et al., 2021). This conceptual connection is currently missing from the paper.

- Several aspects of the main experiments in Section 4.1 could be improved for clarity and completeness:

  - It is unclear which version of SGShift is used in the experiments. Does the implementation include the absorption term, or does it correspond to the knockoff variant? An ablation study illustrating the effect of these different design choices would substantially strengthen the paper.

  - In Table 2 (dataset statistics), the target domain appears to have a sufficient number of samples relative to the feature dimensionality. Given this, it is unclear why baseline methods perform worse, especially since the main critique of plug-in approaches (line 150) was the potential error from training a separate model on the target domain.

   - Please include standard deviations or standard errors in the Table 1 results to improve the statistical rigor and interpretability of the findings.

   - Why does SGShfit continue to perform well even for the dense simulation case? There should be more discussion around this as I would expect the results to be worse since the sparsity assumption is violated.

   -  The observation that matched accuracy is sometimes lower than the unmatched case in Table 1 is counterintuitive and warrants explanation.

- The authors should also include experiments with higher-dimensional feature spaces, as the current setup (maximum of 64 features for the Support2 dataset) is relatively low-dimensional. Even if suitable real-world healthcare datasets are not available, a synthetic high-dimensional dataset could serve to demonstrate the scalability and robustness of the proposed method.

**Questions:**

Please refer to the weaknesses section above for my main questions.

Minor suggestions

- The citation for the WhyShift framework (line 146) is missing,

---

> ### Author Response · Authors · 2025-11-21
> **Response to Reviewer yhDc (1/3)**
>
> Thank you for the accurate summary of our method and for highlighting its rigor and novelty. We summarize each review comment and provide a point-by-point response below. The manuscript has been revised accordingly, incorporating 6 new analyses and 1 update to the main table, and 6 new supplementary figures/tables. This includes 3 new simulation settings, ablation studies, sparsity validation, and covariate shift analysis.
>
>
> > The authors should cite relevant work on causal representation learning to properly justify the sparsity assumption. This assumption is well established in the causality literature through the concept of sparse mechanism shift, which posits that under the correct causal factorization, only a small subset of mechanisms is expected to vary across domains (Schölkopf et al., 2021). This conceptual connection is currently missing from the paper.
>
> Thank you for the suggestion. The concept of sparse mechanism shift in causal representation learning is indeed relevant and well justifies the sparsity assumption. We have added lines 60-61 and lines 136-139 making this connection.
>
> "Just as sparsity is an effective principle for learning predictive models in many settings due to sparse mechanism shift (Schölkopf et al., 2021)"
>
> "For example, this may be the case if specific nodes in a causal process generating the data are intervened on, as is the premise for several previous models of distribution shift (Chen et al., 2022) as well as the concept of sparse mechanism shift in causal representation learning (Schölkopf et al., 2021)."
>
> > It is unclear which version of SGShift is used in the experiments. Does the implementation include the absorption term, or does it correspond to the knockoff variant? An ablation study illustrating the effect of these different design choices would substantially strengthen the paper.
>
> Thank you for the comment, we will provide clarification. In the feature selection tasks, we intend SGShift-K to be the primary tool, and when assessing recovery of model performance, we use naive SGShift or SGShift-A if the model is misfit. We have made this more clear in the main text and figure captions by explicitly mentioning which one is used for each task.
>
> Additionally, we have added ablation studies of feature selection to Appendix H. See results below. We see SGShift-K is indeed the most powerful in identifying shifted features.
>
> Sparse:
> | Dataset              | Group     | SGShift | SGShift-A | SGShift-K |
> | -------------------- | --------- | ------- | --------- | --------- |
> | SUPPORT2             | Matched   | 0.92    | 0.94      | 0.96      |
> | SUPPORT2             | Unmatched | 0.86    | 0.88      | 0.95      |
> | Diabetes Readmission | Matched   | 0.80    | 0.81      | 0.90      |
> | Diabetes Readmission | Unmatched | 0.79    | 0.81      | 0.86      |
> | COVID-19             | Matched   | 0.86    | 0.88      | 0.99      |
> | COVID-19             | Unmatched | 0.85    | 0.8       | 0.97      |
>
> Dense:
> | Dataset              | Group     | SGShift | SGShift-A | SGShift-K |
> | -------------------- | --------- | ------- | --------- | --------- |
> | SUPPORT2             | Matched   | 0.89    | 0.89      | 0.91      |
> | SUPPORT2             | Unmatched | 0.88    | 0.89      | 0.92      |
> | Diabetes Readmission | Matched   | 0.71    | 0.72      | 0.85      |
> | Diabetes Readmission | Unmatched | 0.72    | 0.71      | 0.78      |
> | COVID-19             | Matched   | 0.80    | 0.83      | 0.93      |
> | COVID-19             | Unmatched | 0.76    | 0.76      | 0.91      |

---

> > ### Author Response · Authors · 2025-11-21
> > **Response to Reviewer yhDc (2/3)**
> >
> > > In Table 2 (dataset statistics), the target domain appears to have a sufficient number of samples relative to the feature dimensionality. Given this, it is unclear why baseline methods perform worse, especially since the main critique of plug-in approaches (line 150) was the potential error from training a separate model on the target domain.
> >
> > We thank the reviewer for this insightful comment. While the target domain has a reasonable sample size relative to the feature dimension, plug-in methods must solve a harder problem than single-domain prediction: they must estimate both $\mathbb{E}_S[y|X]$ and $\mathbb{E}_T[y|X]$, and then take their difference. Any estimation noise or modeling bias in either model is amplified in $\hat f_T(X) - \hat f_S(X)$, which both increases variance and destroys the sparsity pattern of the true $\Delta(X)$ by spreading mass over many correlated features. As a result, even with sufficient data for accurate prediction in each domain, recovering a sparse and interpretable $\Delta(X)$ from the noisy difference is challenging. In contrast, SGShift learns a sparse correction to a single source model directly on the target domain, estimating far fewer parameters from target data and preserving the intended sparse structure, likely why it performs strongly in our experiments. We've added more discussion at Lines 153-156.
> >
> > "However, this plugin approach risks an accumulation of errors, particularly when we are interested in recovering structure related to sparsity: given noisy approximations to the two conditional expectations, the difference between $\widehat{\mathbb{E}}_S[y|X]$ and $\widehat{\mathbb{E}}_T[y|X]$ will not necessarily display the same sparsity pattern as $\Delta$ (as we observe experimentally)."
> >
> > > Please include standard deviations or standard errors in the Table 1 results to improve the statistical rigor and interpretability of the findings.
> >
> > Thank you for the comment. We have added 95% CI's aggregated across model configurations to Table 1 as per your suggestion.
> >
> > > Why does SGShfit continue to perform well even for the dense simulation case? There should be more discussion around this as I would expect the results to be worse since the sparsity assumption is violated.
> >
> > Thank you for raising this point. SGShift indeed performs well in the dense simulation because it effectively acts as a regularized feature-ranking procedure: the sparsity-inducing penalty prioritizes features with larger shift magnitude, so with an appropriate penalty level it can still capture most of the signal even when many coefficients are nonzero. This explains why performance degrades only mildly in the dense setting, and we have added a sentence discussing this at Lines 318-320.
> >
> > "While perhaps counterintuitive given the sparsity assumption, SGShift likely performs well as it effectively acts a a regularized feature-ranking procedure, and can still capture most of the signal even when most shift coefficients are nonzero."
> >
> > > The observation that matched accuracy is sometimes lower than the unmatched case in Table 1 is counterintuitive and warrants explanation.
> >
> > Thank you for the comment. We agree that this behavior is counterintuitive. We note, however, that SGShift does not exhibit this pattern: in Table 1, matched performance is consistently stronger than unmatched for SGShift in all scenarios considered. The effect appears mainly for plug-in baselines such as Diff. A plausible explanation is that, even when the model class is matched, the independently trained source and target models can end up with substantially different parameterizations due to optimization and regularization under distribution shift. Their difference can then be noisy and unstable, so that allowing different model classes ("unmatched") sometimes yields a better approximation to the shift by chance.

---

> ### Author Response · Authors · 2025-11-21
> **Response to Reviewer yhDc (3/3)**
>
> > The authors should also include experiments with higher-dimensional feature spaces, as the current setup (maximum of 64 features for the Support2 dataset) is relatively low-dimensional. Even if suitable real-world healthcare datasets are not available, a synthetic high-dimensional dataset could serve to demonstrate the scalability and robustness of the proposed method.
>
> Thank you for the comment. We have incorporated such experiments, simulated high dimensional data with 1000 samples and 500, 200, and 100 features, for each of the 16 model configurations in the original paper. In each case, 20% of the features are randomly selected to induce concept shift. Results are reported below and added to Appendix L. SGShift-K maintains strong performance (all AUC > 0.89) even as the number of features is half the number of samples where other methods lose performance.
>
> 500 Features
> | Group     | Diff | WhyShift | SHAP | SGShift-K |
> | --------- | ---- | -------- | ---- | --------- |
> | Matched   | 0.57 | 0.50     | 0.61 | 0.92      |
> | Unmatched | 0.57 | 0.51     | 0.62 | 0.89      |
>
> 200 Features
> | Group     | Diff | WhyShift | SHAP | SGShift-K |
> | --------- | ---- | -------- | ---- | --------- |
> | Matched   | 0.87 | 0.53     | 0.88 | 0.99      |
> | Unmatched | 0.86 | 0.53     | 0.86 | 0.97      |
>
> 100 Features
> | Group     | Diff | WhyShift | SHAP | SGShift-K |
> | --------- | ---- | -------- | ---- | --------- |
> | Matched   | 0.93 | 0.57     | 0.93 | 1.00      |
> | Unmatched | 0.94 | 0.57     | 0.92 | 1.00      |
>
> Together with reviewer BPYY's suggestions, our work now features 8 total simulation settings (sparse, dense, sample size, global, interaction, high dimensional, highly correlated, and noisy shifts), which we believe constitutes very comprehensive experimental coverage.
>
> > The citation for the WhyShift framework (line 146) is missing,
>
> Thank you for the comment, we have added the missing citation.
>
> We believe that we have successfully addressed your 2 major comments and strongly addressed your requested clarifications. If you feel we have adequately addressed your concerns, we would be grateful if you would you consider updating your evaluation. Thank you again for your time spent reviewing and please let us know if you have any further questions.

---

### Official Review · Reviewer_GD7v · 2025-11-13

**Soundness:** 2
**Presentation:** 1
**Contribution:** 2
**Rating:** 2
**Confidence:** 4

**Summary:**

The paper proposes a method that tackles the problem of concept shift in tabular machine learning tasks, explaining it and rectifying predictor performance degradation caused by it (SGShift). Instead of separately modelling the source and target domains, SGShift learns a sparse corrective term on top of a source-trained model, identifying a small set of features that explain performance degradation under the concept shift. For this, it leverages $l_1$ sparsity inducing regularisation, and knockoffs to avoid over-selection of features.
Experiments are conducted on (semi)synthetic and real healthcare/genetics datasets. On the synthetic data, SGShift is compared to 3 baselines (Diff, WhyShift, SHAP) and is superior than those at recovering shifted features (AUC > 0.9).
The real data experiments illustrate how SGShift explains concept shift in practice and how it can recover performance based on its prediction correction term that only uses very few features.

**Strengths:**

- The paper tackles a relevant and well-scoped problem (concept shift in tabular data with limited target data available)
- The key idea (and potentially contribution) of the paper is simple and easily understandable (the idea of the sparsity of concept shift), and straightforwardly addressed with a simple and easy to implement method.

**Weaknesses:**

Major:
- The paper is unclear at many places and as such hard to correctly interpret or assess (see questions), in particular also as to assessing the strength of the empirical results
- More empirical results on real world data would be needed to understand if the key idea and contribution is really relevant (i.e. if sparsity of concept shift is really a commonly occurring thing in practice)
- Empirical results in Section 4.2 are not compared to any baseline
- The effects of the variants introduced in Section 3.2 and 3.3 (SGShift-A and SGShift-K) are not clear, as the empirical results only contain SGShift (vanilla). Without adding them to the experiment section there is not much point introducing them.


Minor:
- For clarity, I would suggest introducing SGShift-A and -K not just in the subsection titles but also in the text itself as readers sometimes don’t read section titles carefully.
- It would be helpful for the reader to add 1-2 sentences at the end of Section 3 that explain in plain English what the results in Theorem 3.2 mean.
- Figure 1: Please use vector graphics for legend, looks very blurry right now

**Questions:**

- Eq (1): why is $g$ introduced here? It is never used again in the paper afterwards
- Eq (3): what is $\phi_S^T$? The same as $\phi(X_T)^T$? This change in notation should be explained somewhere.
- Line 215: what is $\hat{f}$?
- Line 235: What is $n_T$? Needs to be introduced.
- Lines 234, 235 and 236: What to the modified $\leq$ and $=$ signs stand for? Please define this notation.
- Line 244: what is $\hat{A}^{[b]}$? Where is it defined?
- Line 247: what is $\hat{A}$? Where is it defined?
- Line 268 - 270: is it the features that are shifted (i.e. $P(X)$) or their relationship to the labels $P(Y|X)$? In the text it sounds like it is the features which does not correspond to the problem of concept shift being tackled in the paper though?
- Section 4.1: what regularisation strength $\lambda$ was used for the sparse simulation and dense simulation respectively?
- Section 4.2 healthcare experiment: What is the source and the target data here? How many samples from the target data are being used?
- Section 4.2 genetics experiment: How many samples from the target data are being used? What AUC could be recovered on the Asians with SGShift here?

---

> ### Author Response · Authors · 2025-11-21
> **Response to Reviewer GD7v (1/3)**
>
> Thank you for the thoughtful summary of our method and for highlighting its strong performance in a well-scoped task. We summarize each review comment and provide a point-by-point response below. The manuscript has been revised accordingly, incorporating 6 new analyses and 1 update to the main table, and 6 new supplementary figures/tables. This includes 3 new simulation settings, ablation studies, sparsity validation, and covariate shift analysis.
>
> > More empirical results on real world data would be needed to understand if the key idea and contribution is really relevant (i.e. if sparsity of concept shift is really a commonly occurring thing in practice)
>
> Thank you for the comment. We provide 2 lines of evidence supporting SGShift's utility given the sparsity assumption.
>
> 1. We perform additional performance recovery experiments on datasets with known concept shift from WhyShift's datasets. We train models across 31 state-state pairs in the ACS income datasets, and in 87 cases of concept shift, 78 of these can have model performance completely recovered by less than 1/3 of the total features, 42 of which require less than 10% of features. The remaining 9 can be recovered with less than 50% of features. Including the 16 models considered in Figure 2A, this becomes 94/103 cases of concept shift recoverable with 1/3 the total feature set. We've added this analysis to Appendix O.
> 2. SGShift is robust to the assumption of sparsity. As in Table 1, SGShift still achieves high AUC (avg. AUC=0.9) in detecting shifted features. SGShift's adaptation of sparse regression is still a useful tool for identifying shifted features in less sparse settings.
>
> This assumption is also tied into the causality literature about sparse mechanism shift, see Reviewer yhDC's comment: "This assumption is well established in the causality literature through the concept of sparse mechanism shift, which posits that under the correct causal factorization, only a small subset of mechanisms is expected to vary across domains (Schölkopf et al., 2021)."
>
>
> > Empirical results in Section 4.2 are not compared to any baseline
>
> Thank you for the comment. Our goal is to identify the set of shifted features, not maximize model performance on the target distribution. Figure 2 is intended to serve as validation that concept shift can indeed be sparse. That is, the important feature of the plot is that performance has a sharp U shape (indicating sparsity of the difference), not whether the curve could be made still sharper by another method. We acknowledge that existing work does aim to address the challenge of performance drop under concept shift. Using SGShift's learned features to maximize model performance is an interesting future direction but we don't make any claims of improvement at that task in this paper.
>
> > The effects of the variants introduced in Section 3.2 and 3.3 (SGShift-A and SGShift-K) are not clear, as the empirical results only contain SGShift (vanilla). Without adding them to the experiment section there is not much point introducing them.
>
> Thank you for the comment, we will provide clarification. In the feature selection tasks, we intend SGShift-K to be the primary tool, and when assessing recovery of model performance, we use naive SGShift, or SGShift-A if the model is misfit. We have made this more clear in the main text and figure captions by explicitly mentioning which one is used for each task.
>
> Additionally, we have added ablation studies of feature selection to Appendix H. See results below. We see SGShift-K is indeed the most powerful in identifying shifted features.
>
> Sparse:
> | Dataset              | Group     | SGShift | SGShift-A | SGShift-K |
> | -------------------- | --------- | ------- | --------- | --------- |
> | SUPPORT2             | Matched   | 0.92    | 0.94      | 0.96      |
> | SUPPORT2             | Unmatched | 0.86    | 0.88      | 0.95      |
> | Diabetes Readmission | Matched   | 0.80    | 0.81      | 0.90      |
> | Diabetes Readmission | Unmatched | 0.79    | 0.81      | 0.86      |
> | COVID-19             | Matched   | 0.86    | 0.88      | 0.99      |
> | COVID-19             | Unmatched | 0.85    | 0.8       | 0.97      |
>
> Dense:
> | Dataset              | Group     | SGShift | SGShift-A | SGShift-K |
> | -------------------- | --------- | ------- | --------- | --------- |
> | SUPPORT2             | Matched   | 0.89    | 0.89      | 0.91      |
> | SUPPORT2             | Unmatched | 0.88    | 0.89      | 0.92      |
> | Diabetes Readmission | Matched   | 0.71    | 0.72      | 0.85      |
> | Diabetes Readmission | Unmatched | 0.72    | 0.71      | 0.78      |
> | COVID-19             | Matched   | 0.80    | 0.83      | 0.93      |
> | COVID-19             | Unmatched | 0.76    | 0.76      | 0.91      |

---

> > ### Author Response · Authors · 2025-11-21
> > **Response to Reviewer GD7v (2/3)**
> >
> > > For clarity, I would suggest introducing SGShift-A and -K not just in the subsection titles but also in the text itself as readers sometimes don't read section titles carefully.
> >
> > Thank you for the comment, we have now introduced them in the text at lines 285-287.
> >
> > "In feature selection tasks, we primarily use SGShift-K with knockoffs, and in model performance recovery we use naive SGShift and SGShift-A with absorption. "
> >
> > > It would be helpful for the reader to add 1-2 sentences at the end of Section 3 that explain in plain English what the results in Theorem 3.2 mean.
> >
> > Thank you for the comment, we have added intuition of Theorem 3.2 at lines 267-268.
> >
> > "Theorem 3.2 guarantees both per family error rate (PFER) and false discovery rate (FDR) control under proper parameter selection"
> >
> > > Figure 1: Please use vector graphics for legend, looks very blurry right now
> >
> > Thank you for the comment, we have made the figure legend higher quality.
> >
> > > Eq (1): why is $g$ introduced here? It is never used again in the paper afterwards
> >
> > Thank you for the comment. $g$ is introduced at line 134 as the link function used for modeling $\Delta(X)$, used in the proof of FDR control in Appendix E, and for inducing shifted features in the simulations (Line 278).
> >
> > > Eq (3): what is $\phi_S^T$? The same as $\phi(X_T)^T$? This change in notation should be explained somewhere.
> >
> > Thank you for the comment. $\phi^T_S$ is $\phi(X_S)^T$, or the values of the set of basis functions $\phi$ for $X$ in the source domain. We've now introduced these explicitly at line 209.
> >
> > "where $\phi_S$ and $\phi_T$ refer to the values of  basis functions in source and target domains,"
> >
> > > Line 215: what is $\hat{f}$?
> >
> > Thank you for the comment. As per the original version's line 215, this is the source predictor, specifically, the trained source model. We've updated this line to be more clear, now one line 225.
> >
> > "predictive model trained on the source domain"
> >
> > > Line 235: What is $n_T$? Needs to be introduced.
> >
> > Thank you for the comment. $n_T$ is the number of samples in the source domain, introduced at line 126. We've now clarified this in line 127.
> >
> > "where $n_S$ and $n_T$ are the number of samples in source and target domain"
> >
> > > Lines 234, 235 and 236: What to the modified $\leq$ and $=c$ signs stand for? Please define this notation.
> >
> > Thank you for the comment. These are standard notations for "asymptotically bounded above up to a constant factor" and "asymptotically equal to". We added their explicit definitions to lines 248-249.
> >
> > "Here, $\lesssim$ means asymptotically bounded above up to a constant factor, and $\asymp$ means asymptotically the same order up to constant factors."
> >
> > > Line 244: what is $\hat{A}^{[b]}$? Where is it defined?
> >
> > Thank you for the comment. $A^c$ is the set of features with zero coefficient in the true data distribution and B is the number of knockoff samples (Lines 244-245). $\hat{A}^{[b]}$ is the predicted set of features with nonzero coefficient for knockoff sample $b$. We've added a line clarifying this at Line 256.
> >
> > "$\hat{A}^{[b]}$ is the estimated $A$ for $b$th knockoff repeat, and $\hat{A}(\pi)$ is the selection across all repeats under stability threshold $\pi$"
> >
> > > Line 247: what is $\hat{A}$? Where is it defined?
> >
> > See above comment.
> >
> > > Line 268 - 270: is it the features that are shifted (i.e. $P(X)$) or their relationship to the labels $P(Y|X)$? In the text it sounds like it is the features which does not correspond to the problem of concept shift being tackled in the paper though?
> >
> > Thank you for the comment. Our goal is only to identify features with shifted relationship to the labels. We are abbreviating "features with shifted relationship to the labels" as "shifted features" throughout the paper. We've added a line clarifying this at Lines 70-71.
> >
> > "identifying concept shifted features (referred to as shifted features throughout)"

---

> ### Author Response · Authors · 2025-11-21
> **Response to Reviewer GD7v (3/3)**
>
> > Section 4.1: what regularisation strength $\lambda$ was used for the sparse simulation and dense simulation respectively?
>
> Thank you for the comment. We vary the regularization from 0.0001 to 100 to obtain a ranking of when shifted features enter the model. We've clarified this at lines 287-288.
>
> "SGShift's feature ranking is obtained by varying the penalty parameter from 0.0001 to 100 to measure AUC and recall."
>
> > Section 4.2 healthcare experiment: What is the source and the target data here? How many samples from the target data are being used?
>
> Thank you for the comment. We are using the same source and target data as was used to generate simulations (Table 2). We've added a line clarifying this line 400.
>
> "(data split in Table 2)"
>
> > Section 4.2 genetics experiment: How many samples from the target data are being used? What AUC could be recovered on the Asians with SGShift here?
>
> Thank you for the comment. As per line 409, we use data from "107 healthy and Lupus-affected Asians". An AUC of 1.0 could be recovered, we've added this to the main text at line 414-415.
>
> "and completely recovers the performance drop"
>
> We believe that we have successfully addressed your 3 major comments, particularly about the point that we are indeed using different SGShift variants, and strongly addressed your requested clarifications. If you feel we have adequately addressed your concerns, we would be grateful if you would you consider updating your evaluation. Thank you again for your time spent reviewing and please let us know if you have any further questions.

---

### Author Response · Authors · 2025-12-02
**Final Overall Response**

Thank you for your efforts in handling submissions during the recent unexpected situation. To facilitate your evaluation, we provide a concise summary of the review history and our rebuttal progress.

During the rebuttal period, we conducted substantial new analyses, including:

(i) 3 additional requested simulation settings (high-dimensional, high feature correlation, low signal-to-noise) which SGShift achieves very strong and top performance in for a total of 8 simulation settings;

(ii) ablation studies confirming the effectiveness of introducing knockoffs into SGShift’s feature selection, and

(iii) confirmation of the sparsity of real world concept shift with analysis of 87 further real world pairs of source-target distributions from distribution shift benchmarks, finding 78 of these cases can have concept shift completely recovered with less than ⅓ of all the features.

The manuscript has been revised accordingly, incorporating 6 new analyses and 1 update to the main table, and 6 new supplementary figures/tables.

Our submission had an initial average score of 4. While we were not able to receive any responses due to the shortened rebuttal period, we have thoroughly responded to all reviewers’ concerns and questions and therefore expect that the final average score would have been significantly higher had reviewers been allowed to complete their updates. Reviewers consistently acknowledged our paper as a well-presented method (spce, yhDc, BPYY) with strong performance in a relevant task (GD7V, yhDc, spce) and were onboard with the key idea of the paper (GD7V, yhDc, spce). Reviewer requests were largely for clarification or additional experiments to corroborate different aspects of the findings which we have provided in our rebuttal response.

We’d also like to summarize our key technical contributions. SGShift aims to detect the features contributing to conditional distribution shift $p(y|X)$ between domains. This direction is less explored in related work, where usually the primary aim is performance recovery, rather than feature identification. We frame the problem as learning a sparse correction term between source and target models and adapt powerful tools for feature selection to this framing. Across 8 simulation settings, including sparse, dense, global, interaction, sample size, high dimensional, highly correlated, and low signal-to-noise ratio, SGShift is able to achieve top performance at detecting shifted features compared to baselines, often drastically so with AUC > 0.2 higher than the closest baseline.

Below we summarize our responses to each reviewer:

**Reviewer spce (score 6).**

The core question raised by the reviewer was if the real world concept shift identified by SGShift could be attributable to covariate shift instead, and we presented experiments with inverse propensity weighting to account for covariate shift and found that covariate shift correction alone did not recover model performance, while SGShift’s concept shift correction could, validating our findings. While we did not receive a response due to the shortened discussion period, we believe the reviewer would be satisfied with our rebuttal.

**Reviewer yhDc (score 4).**

The core questions raised by the reviewer were regarding clarity of the main experiment Section in 4.1, which we have updated the manuscript to clarify, and a request for an additional high dimensional experiment which we have now provided and shown SGShift achieves very strong and top performance in. While we did not receive a response due to the shortened discussion period, we believe the reviewer would be satisfied with our rebuttal.

**Reviewer BPYY (score 4).**

The core questions raised by the reviewer were requests for additional highly correlated and low signal-to-noise simulation settings, which we have now provided and shown SGShift achieves very strong and top performance in, and an additional request for ablations showing the utility of each SGShift variant, which we have provided and shown the utility of our knockoff adaptation at feature selection. While we did not receive a response due to the shortened discussion period, we believe the reviewer would be satisfied with our rebuttal.

**Reviewer GD7v (score 2).**

The core questions raised by the reviewer were requests for clarification, validation of real world sparsity, and for ablations showing the utility of each SGShift variant. We have added all requested clarifications, analysis of 87 further real world pairs of source-target distributions from distribution shift benchmarks, finding 78 of these cases of concept shift are indeed sparse, and provided ablations showing the utility of our knockoff adaptation at feature selection. While we did not receive a response due to the shortened discussion period, we believe the reviewer would be satisfied with our rebuttal.

In summary, we believe we have substantially addressed all reviewer concerns. We appreciate your consideration.

---

### Meta-Review · Area_Chair_bbSY · 2025-12-10

**Summary:**

The reviewers had different appreciations of this work, with one reviewer being more positive while the others leaned towards rejection.

In summary, I have identified some repeated concerns about the paper:
- lack of baseline comparisons in real-world scenarios
- whether concept shift is a significant issue and whether it has been identified in the presented real-world scenarios
- confusion around the different variants of the method and what they are used for

**Reviewer Concerns:**

The authors have provided extensive responses, which I believe have partly addressed the issues.

(1) *lack of baseline comparisons in real-world scenarios*: The argument is that these scenarios aim to show that concept shift is feature-sparse in real-world scenarios and not focused on providing the best achievable performance. While I understand the argument, I don't believe it would have convinced the reviewers, as the U-shape is method-dependent and it would be important to understand how "far" performance on the target would be if we were to correct the features from the proposed method. The authors leave this for future work but I believe the point made is important.

(2) *whether concept shift is a significant issue* : I think the authors misunderstood reviewer GD7v's first question, which directly relates to reviewer's spce. The extra experiments with propensity scores provide an interesting first step, but it is questionable whether the shifts in the real-world scenarios truly represent concept shifts.

(3) *confusion around variants*: I believe the authors have clarified this in their response, although having to use different variants in different cases may limit the applicability of the method in my opinion.

**Reviewer Scores:**

Based on all this context, I don't believe the reviewers would have significantly changed their scores. I am also accounting for the confidence in the reviews, where the only positive review had much lower confidence than the others.

---

### Decision · Program_Chairs · 2026-01-26

Reject